# Strand-specific, high-resolution mapping of modified RNA polymerase II

Laura Milligan[1], Vân A Huynh-Thu[2,3], Clémentine Delan-Forino[1], Alex Tuck[1,4,5], Elisabeth Petfalski[1], Rodrigo Lombraña[6], Guido Sanguinetti[2,***], Grzegorz Kudla[6,**] & David Tollervey[1,*]

## Abstract

Reversible modification of the RNAPII C-terminal domain links transcription with RNA processing and surveillance activities. To better understand this, we mapped the location of RNAPII carrying the five types of CTD phosphorylation on the RNA transcript, providing strand-specific, nucleotide-resolution information, and we used a machine learning-based approach to define RNAPII states. This revealed enrichment of Ser5P, and depletion of Tyr1P, Ser2P, Thr4P, and Ser7P in the transcription start site (TSS) proximal ~150 nt of most genes, with depletion of all modifications close to the poly(A) site. The TSS region also showed elevated RNAPII relative to regions further 3′, with high recruitment of RNA surveillance and termination factors, and correlated with the previously mapped 3′ ends of short, unstable ncRNA transcripts. A hidden Markov model identified distinct modification states associated with initiating, early elongating and later elongating RNAPII. The initiation state was enriched near the TSS of protein-coding genes and persisted throughout exon 1 of intron-containing genes. Notably, unstable ncRNAs apparently failed to transition into the elongation states seen on protein-coding genes.

**Keywords** hidden Markov model; polymerase CTD phosphorylation; transcription; yeast

**Subject Categories** Chromatin, Epigenetics, Genomics & Functional Genomics; Methods & Resources; Transcription

**Mol Syst Biol. (2016) 12: 874**

## Introduction

Eukaryotic RNA polymerase II (RNAPII) is a large, multisubunit complex. The largest RNAPII subunit, termed Rpo21 in yeast, has a catalytic N-terminal domain (NTD) that performs nucleotide polymerization, and a regulatory C-terminal domain (CTD), consisting of multiple heptad (consensus YSPTSPS) repeats (26 in yeast, 52 in humans). Each position in the heptads of the CTD can undergo independent, reversible post-translational modification (PTM) (reviewed in Heidemann *et al*, 2013). In particular, phosphorylation/dephosphorylation of Tyr1, Ser2, Thr4, Ser5, and Ser7 can regulate interactions with multiple transcription and RNA processing factors (reviewed in Buratowski, 2009; Hsin & Manley, 2012). Two recent studies addressed the mapping of these phosphorylation sites of the CTD in a heptad-specific manner (Schüller *et al*, 2016; Suh *et al*, 2016). Both studies indicate that Ser5P and Ser2P are the major modifications; however, other phosphorylation sites are clearly also functionally important (Egloff *et al*, 2007; Hsin *et al*, 2011; Mayer *et al*, 2012; Rosonina *et al*, 2014). The RNAPII CTD shows dynamic changes in PTMs as the polymerase traverses protein-coding genes in yeast (see, e.g., Mayer *et al*, 2010), which can be detected using a set of antibodies that each recognize a specific phosphorylation (reviewed in Eick & Geyer, 2013; Heidemann *et al*, 2013). Such antibodies have been used to assess the distribution of the different modifications across individual transcription units in ChIP analyses, or across many transcription units in ChIP-Seq. These analyses have been highly informative and have played important roles in our current understanding of eukaryotic gene expression. However, ChIP and ChIP-Seq map RNAPII by formaldehyde crosslinking in sonicated chromatin, so the transcribed strand cannot be identified and spatial resolution is limited by the fragment sizes—typically around 200–300 nt. In contrast, recent analyses of RNA-binding proteins have used UV crosslinking and sequence small RNA fragments, giving a high degree of spatial resolution (reviewed in Darnell, 2010). Here, we have applied *in vivo* UV crosslinking and analysis of cDNA (CRAC) (Granneman *et al*, 2009) to generate a high-resolution, strand-specific map of the location of transcribing RNAPII relative to the nascent transcript. We also incorporated an additional immunoprecipitation step to separate modified forms of

1 Wellcome Trust Centre for Cell Biology, University of Edinburgh, Edinburgh, UK
2 School of Informatics, University of Edinburgh, Edinburgh, UK
3 Department of Electrical Engineering and Computer Science, University of Liège, Liège, Belgium
4 Friedrich Miescher Institute for Biomedical Research, Basel, Switzerland
5 European Molecular Biology Laboratory, European Bioinformatics Institute (EMBL-EBI), Wellcome Trust Genome Campus, Cambridge, UK
6 MRC Human Genetics Unit, IGMM, University of Edinburgh, Edinburgh, UK
 *Corresponding author. Tel: +44 131 650 7092; E-mail: d.tollervey@ed.ac.uk
 **Corresponding author. Tel: +44 131 651 8628; E-mail: gkudla@gmail.com
 ***Corresponding author. Tel: +44 131 650 5136; E-mail: gsanguin@inf.ed.ac.uk

RNAPII (modification CRAC; mCRAC) and map these across the transcriptome. A related approach was recently reported to identify the locations of Ser5P- and Ser2P-modified RNAPII in human cells (Nojima *et al*, 2015).

Extracting biological insights from large and complex transcriptomic or genomic datasets remain challenging. To address this problem, modeling approaches can help identify significant common features across genes and between gene classes in metagene analyses based on sequence analyses. We have therefore used machine learning and probabilistic modeling to segment the genome using a hidden Markov model (HMM). The HMM explains the mCRAC data by postulating an underlying, finite set of hidden RNAPII states, each characterized by a noisy signature of PTM log-enrichments (emissions). RNAPII can transition between different states stochastically; however, the model assumptions enforce spatial coherence in the state distribution, effectively yielding a partitioning of the genome according to different HMM states. See Eddy (2004) for an accessible overview of the use of HMMs to analyze biological data. HMMs have recently been used to define features of yeast transcription units and the transcription initiation–elongation transition (de Boer *et al*, 2014; Zacher *et al*, 2014).

Like other eukaryotes, yeast transcribes large numbers of ncRNAs. These include the cryptic unstable transcripts (CUTs), which are normally very rapidly degraded (Wyers *et al*, 2005), and the stable untranslated transcripts (SUTs) a class of generally nuclear retained, but more stable ncRNAs (Xu *et al*, 2009). CUTs are strongly stabilized by mutations in the exosome nuclease complex, the Trf4/5-Air1/2-Mtr4 (TRAMP) nuclear polyadenylase complex, a major exosome cofactor, or the Nrd1 and Nab3 components of the Nrd1/Nab3/Sen1 (NNS) complex that terminates transcription and targets the nascent transcript to TRAMP and the exosome (Arigo *et al*, 2006; Heo *et al*, 2013; Schulz *et al*, 2013; Tudek *et al*, 2014; Webb *et al*, 2014; Fasken *et al*, 2015). These observations strongly support the model that CUTs are transcriptionally terminated by NNS, with the released transcripts being targeted to the TRAMP/exosome complex for rapid degradation. The NNS complex can associate with Ser5 phosphorylated RNAPII CTD via Nrd1 (Vasiljeva *et al*, 2008) and with the histone H3, lysine 4 trimethylation (H3K4me3) present on the promoter-proximal nucleosome(s) (Terzi *et al*, 2011), and Nab3 can also directly recruit the Rrp6 component of the exosome (Fasken *et al*, 2015). However, features that discriminate between mRNAs and ncRNA transcripts to confer their different fates are not fully understood. Consensus *in vitro* binding sites for Nrd1 and Nab3 are depleted in protein-coding genes relative to the total genome or ncRNAs (Schulz *et al*, 2013). However, while Nrd1 and Nab3 show a strong 5′ bias, they also bind throughout protein-coding genes (Webb *et al*, 2014) and individual *in vivo* Nrd1 and Nab3 binding sites frequently lack these motifs (Jamonnak *et al*, 2011; Wlotzka *et al*, 2011), indicating that these are not the sole determinants of NNS function *in vivo*.

Systematic analyses revealed that the pre-mRNA-binding proteins Hrp1 and Nab2 have dual roles, promoting both pre-mRNA cleavage and polyadenylation, and degradation of CUT ncRNAs (Tuck & Tollervey, 2013). Surprisingly, on protein-coding transcripts, Hrp1 and Nab2 were most abundant close to the transcription start site, and this was also found to be the case for the nuclear surveillance factor Mtr4. These observations suggested that short ncRNAs might be generated from the 5′ regions of many or all protein-coding genes in yeast.

Here, we report that the combination of high-resolution, strand-specific mapping of RNAPII phosphorylation states across the transcriptome with an HMM allowed the segmentation of the genome based on RNAPII state changes. This segmentation can readily be compared to any other feature for which genome-wide data exist. We use this approach to reveal systematic differences between RNAPII on intron-containing and intronless transcripts, and between mRNAs and lncRNAs. We propose that close to the transcription start site (TSS) RNAPII is normally "surveillance ready". The early post-initiation state for RNAPII transcripts is prone to termination, with degradation of the released transcript. On genes encoding unstable CUT ncRNAs, RNAPII remains in this state. We postulate that, in consequence, the surveillance system does not need to specifically identify thousands of different ncRNA species.

## Results

### Rpo21 CRAC recovers nascent RNAPII transcripts

To assess the distribution of total RNAPII along the yeast genome, we applied CRAC to Rpo21, the large subunit of the RNAPII complex. HTP-tagged Rpo21 was expressed from the endogenous locus with a His6-TEV-protein A tag (Rpo21-HTP, Fig EV1). Rpo21-HTP was the only source of Rpo21 and growth of the resulting strains was indistinguishable from the parental wild type (BY4741), indicating that the tagged construct is functional. Rpo21 was cross-linked to RNA by UV irradiation at 254 nm for 1 min in actively growing cells and purified using a multistep, tandem-affinity protocol, followed by gel purification (Figs 1A and EV1). The protocol includes two highly denaturing steps (incubation and nickel column binding in 4 M guanidinium–HCl and SDS gel electrophoresis) ensuring very high enrichment for covalently associated RNAs. RNA fragments crosslinked to Rpo21 were amplified and sequenced, yielding 1.5 million unique reads that could be mapped to the genome. After quality filtering and removal of PCR duplicates, more than 90% of these sequences were mapped to genes transcribed by RNAPII, including mRNAs, several lncRNAs classes (CUTs, SUTs, NUTs, XUTs, and antisense transcripts), small nucleolar RNAs (snoRNAs), and small nuclear RNAs (snRNAs), indicating specific recovery of RNAPII transcripts (Fig 1B).

Fig 1C shows the RNAPII binding profile on all protein-coding genes, aligned by the transcription start site (TSS) and arranged by transcript length. Robust RNAPII binding was found on the majority of mRNAs, suggesting that most mRNAs are expressed and detected by CRAC. This analysis showed that high signals on the sense strand were not accompanied by antisense signals, confirming the strand specificity of the CRAC technique. The distribution of RNAPII across selected individual genes is shown in Dataset EV1. Inspection of the total RNAPII signals on the plus and minus strands in panels A–C shows the high strand specificity of the CRAC data. Dataset EV1D shows the *PHO84* gene, which has a well-characterized, functionally important antisense transcript (Camblong *et al*, 2007). Sense transcription and antisense transcription are clearly resolved in the CRAC data, but would be conflated in ChIP analyses.

Pre-mRNA splicing in yeast is very rapid and largely takes place on the nascent transcript (Moehle *et al*, 2014). To calculate relative abundances of spliced and unspliced reads, we mapped the reads to

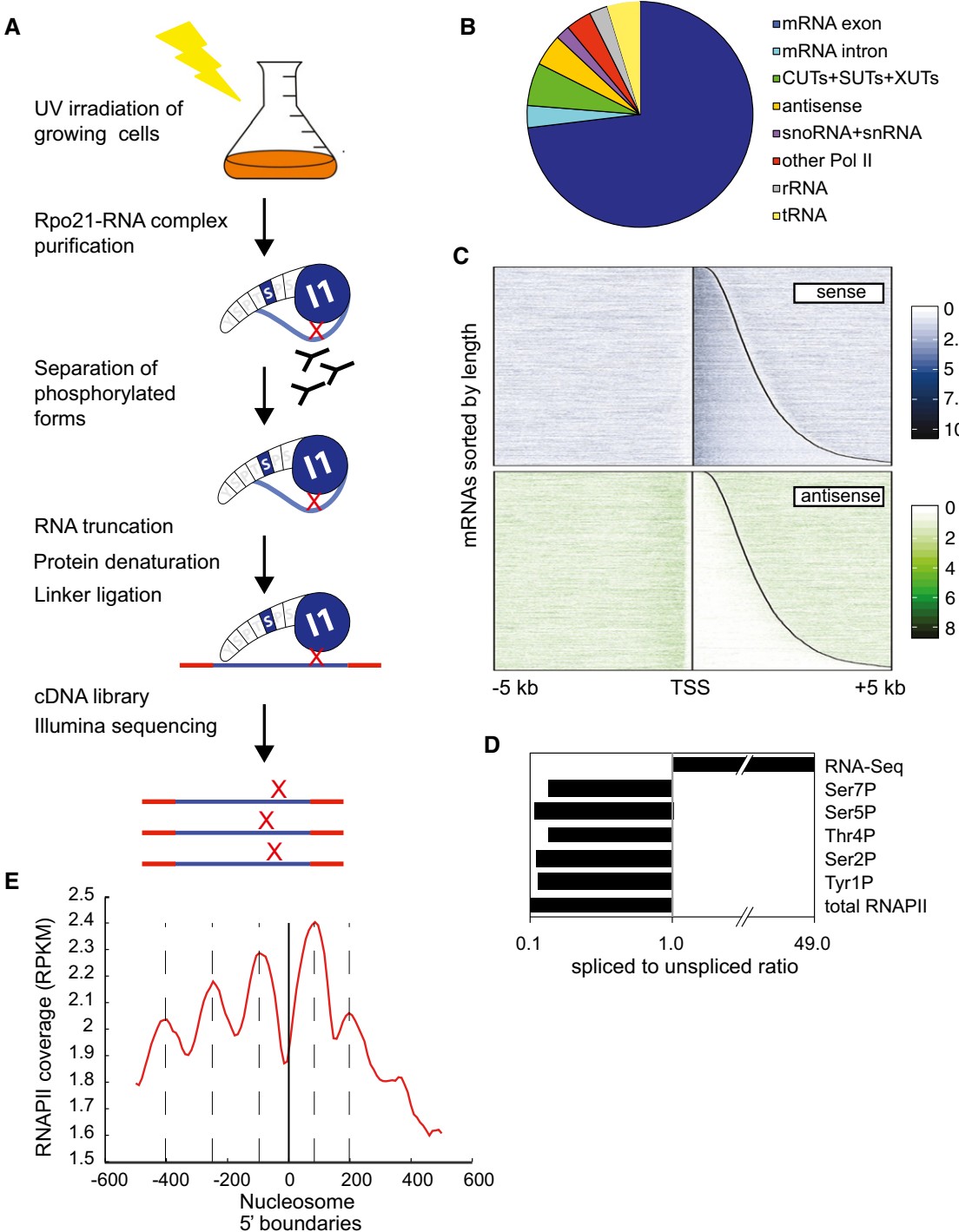

**Figure 1. RNAPII can be mapped with high-resolution using CRAC.**

A   Outline of the mCRAC protocol. See Fig EV1 for further details. In all figures, the analyses used *S. cerevisiae* strains derived from BY4741.

B   Distribution of RNAPII reads across transcript classes determined by CRAC analyses of Rpo21-HTP.

C   Distribution of RNAPII across protein-coding genes in the sense and antisense orientations. In the upper panel, the vertical line indicates the TSS. The curved line indicates the location of the poly(A). All protein-coding genes are shown in the sense orientation, ordered with the shortest ORF at the top. The lower panel shows reads that are antisense to the same regions.

D   Ratio of spliced to unspliced RNAs in RNAPII-bound RNAs, calculated as the ratio of sequences spanning exon–exon (spliced) relative to intron–exon (unspliced) junctions.

E   Peaks in RNAPII binding correlate with nucleosome positions. The zero point (solid vertical line) is the mapped positions of nucleosome 5′ boundaries (Jiang & Pugh, 2009) across all protein-coding genes. The red line shows the overall RNAPII density with respect to each nucleosome boundary. Dashed lines show locations RNAPII maxima, which show an apparent 150 nt periodicity.

a database containing all known splice junctions in mRNA sequences. The proportion of spliced to unspliced reads was calculated as $(2 \times ExEx)/(ExI+IEx)$, where ExEx is the number of reads spanning exon–exon junctions, ExI is the number of reads spanning exon–intron junctions (5′ splice sites), and IEx is the number of reads spanning intron–exon junctions (3′ splice sites). Pre-mRNAs (ExI+IEx reads) were strongly enriched relative to mature RNAs (ExEx reads), consistent with the recovery of *bona fide* RNAPII-associated, nascent transcripts (Fig 1D).

We noted that the RNAPII distribution was frequently uneven along individual genes. It seemed possible that this reflected changes in RNAPII elongation rates in response to the presence of nucleosomes on the DNA template. The density of RNAPII crosslinking across all protein-coding genes was therefore mapped with respect to nucleosome boundaries (Fig 1E). A striking pattern f RNAPII hit density was observed, with strong 150 nt periodicity. Nucleosome arrays are formed on most yeast genes (Jiang & Pugh, 2009; Weiner *et al*, 2010), and the distribution of RNAPII peaks matches the nucleosome spacing. The locations of nucleosomes on individual genes are shown in Dataset EV1.

These data would be consistent with the model that nucleosomes remain associated with the DNA during transcription, with RNAPII moving more slowly through the center of the nucleosome-bound DNA region than the outside and spacers. Similar conclusions were drawn from the analysis of RNAPII pause sites determined by NET-seq analysis (Churchman & Weissman, 2011).

The RNAPII signal dropped sharply in the 3′ untranslated regions (3′ UTRs) of protein-coding genes. The principal drop in RNAPII density was centered around the translation stop codon (Fig EV2), which in yeast is generally located approximately 50–100 nt upstream of the principal polyadenylation p(A) sites, indicated by the distribution of the poly(A)-binding protein (Pab1) in Fig EV2.

Motif analysis upstream of the poly(A) showed strong enrichment for the canonical polyadenylation signal AWUAAA and the binding site for the polyadenylation factor Hrp1 (Nab4), AUAUAU (Fig EV2) (Tuck & Tollervey, 2013). Similar reduced RNAPII occupancy is visible in NET-Seq data (see figures 2 and 6 in Nojima *et al*, 2015) and is consistent with the formation of a distinct Pol II elongation complex close to the termination site (see below and Mayer *et al*, 2010). Nucleosome analysis (Fig EV2) (Jiang & Pugh, 2009; Weiner *et al*, 2010) showed a peak of nucleosome occupancy immediately upstream of the stop codon, but the significance of this is unclear. In ChIP analyses, increased RNAPII density was observed at the 3′ ends of mRNAs. However, Fig 1C shows that antisense ncRNA transcripts are detected in the 3′ regions of many protein-coding genes, which will contribute to RNAPII ChIP signals.

## The 5′ region of protein-coding genes shows high occupancy by RNAPII and surveillance factors

Analysis of the distribution of RNAPII across protein-coding genes (Fig 1C) indicated systematically higher levels close to the TSS, as previously observed in NET-seq analyses in yeast and human cells (Churchman & Weissman, 2011; Mayer *et al*, 2015; Nojima *et al*, 2015). Filtering the genes for transcripts longer than 500 nt in length confirmed that the RNAPII signal is indeed elevated close to the TSS (Fig 2A). Previous analyses showed that the RNA-binding proteins Hrp1 and Nab2 have dual functions in promoting mRNA 3′ end

formation and transcription termination, and destabilizing CUT ncRNAs (Tuck & Tollervey, 2013). Notably, the distribution of Hrp1 and Nab2, as well as the nuclear surveillance factor Mtr4, was heavily skewed toward the TSS (Fig 2), suggesting the possibility that many protein-coding genes generate short, truncated ncRNAs in addition to the full-length mRNA. CUTs and other rapidly degraded nuclear RNAs are stabilized by loss of the exosome nuclease complex, or its cofactors the TRAMP polyadenylation complex and the NNS transcription termination/surveillance complex. To assess whether binding of the nuclear surveillance system is generally enriched close to the TSS, we compared the distribution of the catalytic components of the nuclear exosome, Rrp6 and Rrp44, as well as its major nuclear cofactors, the TRAMP complex components Trf4 and Air2 (this work), and the NNS transcription termination complex component Nab3 (Holmes *et al*, 2015). Strikingly each of these major nuclear RNA surveillance factors showed a strong 5′ bias in pre-mRNA association, with peak binding within ~150 nt from the TSS (Fig 2B–F). The major RNA export factor Mex67 binds newly synthesized nuclear pre-mRNAs but is dissociated immediately following nuclear export (Segref *et al*, 1997; Lund & Guthrie, 2005), and is depleted from ncRNAs relative to mRNAs (Tuck & Tollervey, 2013). The distribution of Mex67 was strikingly different from the surveillance factors, showing strong depletion close to the TSS (Fig 2G) (Tuck & Tollervey, 2013). These observations are consistent with a substantial degree of transcription termination in the promoter-proximal region of protein-coding genes and, indeed, the 3′ ends of these RNAs can be observed in published RNAseq data (Fig 2H) (Neil *et al*, 2009).

## RNAPII shows distinct features close to the TSS and pA site

To examine transcriptome-wide profiles of RNAPII phosphorylation, we included an additional immunoprecipitation step in the mCRAC procedure (Figs 1A and EV1), which allowed us to map the distribution of Tyr1P, Ser2P, Thr4P, Ser5P, or Ser7P phosphorylation in the CTD of RNAPII (Figs 3A and EV3). To account for unequal Pol II distribution along the genome, phosphorylation enrichment was quantified as log2 (phosphorylated Pol II/total Pol II) (see Materials and Methods). Phosphorylation profiles were remarkably similar among mRNA genes and were independent of gene length (Figs 3A and B, and EV3). Genome-wide locations of Tyr1P, Ser2P, Thr4P, and Ser7P enrichment were strongly correlated with each other and anticorrelated with Ser5P. Ser5P was enriched near mRNA 5′ ends, as expected, but sharply declined further 3′. In contrast, Tyr1p, Ser2P, Thr4P, and Ser7P were all strongly depleted near the 5′ end. Tyr1P and Ser2P were depleted in a narrow window of 0–150 nt downstream from the TSS, whereas Thr4P and Ser7P were depleted in a relatively broader area (0–300 nt from TSS), suggesting that these CTD phosphorylations might be placed in an ordered fashion during transcription.

Alignment of the RNAPII modification profiles by the pA sites gave a different result, with strong depletion of all modification immediately upstream of the pA site (Fig 3B). Dephosphorylation of Tyr1 by Glc7 was previously shown to be important for mRNA 3′ end formation (Schreieck *et al*, 2014), but these findings indicate that other residues are also dephosphorylated in this region. We speculate that this is related to the reduced occupancy observed for total RNAPII in this region in CRAC and NETseq analyses (Fig EV2;

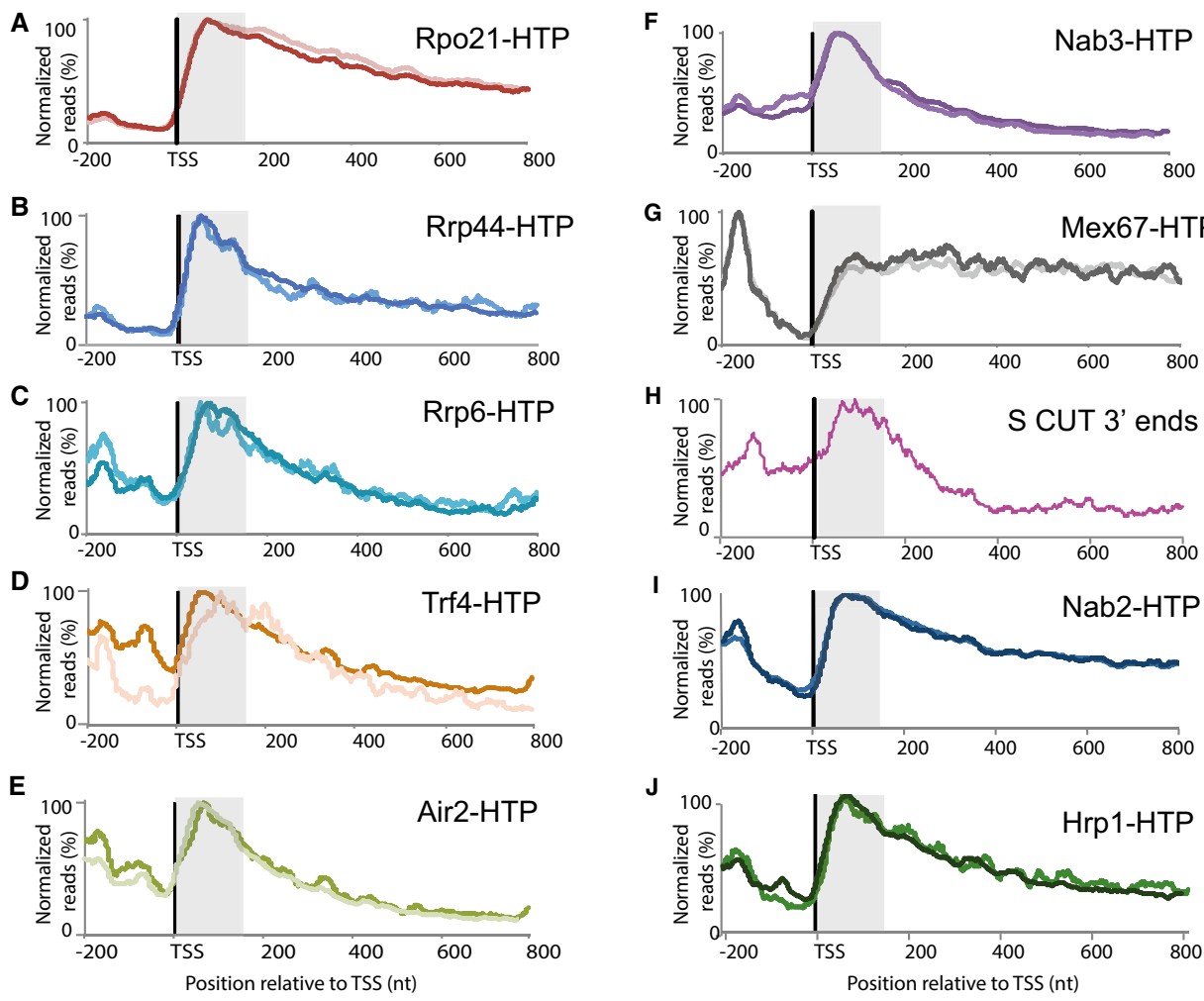

**Figure 2.  Binding by RNA surveillance and degradation factors is strongly enriched close to the TSS on protein-coding genes.**

A–J   Each panel shows the hit density, normalized to the maximal binding value across all mRNA genes longer than 500 nt inside each individual experiment. The two lines in each panel represent results from independent CRAC experiments. The TSS-proximal 150 nt region is shaded. (A) Rpo21 (RNAPII); (B) Rrp44; (C) Rrp6; (D) Trf4; (E) Air2; (F) Nab3; (G) Mex67; (H) Distribution of the 3′ ends of short, promoter-proximal, sense-orientated ncRNA transcripts (S CUTs); (I) Nab2; (J) Hrp1. Sequence data source: (A-E) (this work), (F) (Holmes *et al*, 2015). (G, I, J) (Tuck & Tollervey, 2013), (H) (Neil *et al*, 2009).

Churchman & Weissman, 2011; Mayer *et al*, 2015; Nojima *et al*, 2015).

Sites of interaction of the transcribing polymerase with the nascent transcript were previously mapped by sequencing of protected fragments (NET-seq) in yeast (Churchman & Weissman, 2011) and human cells (Nojima *et al*, 2015). The major NET-seq products in yeast were reported to represent sites of RNAPII transcription pausing. To assess the relationship between RNAPII modification patterns and transcriptional pausing, we aligned the strongest reported pause sites over all protein-coding genes with enrichment for the different RNAPII modifications. Notably higher levels of Ser5 phosphorylation were seen immediately 5′ to the pause site compared to the 3′ flanking region. In contrast, maxima for Tyr1, Ser2, Thr4, and Ser7 phosphorylation were located 3′ to the pause sites (Fig 3C).

In analyzing the distribution of the different modified forms of RNAPII, we expressed the relative abundance at each position as a fraction of the total RNAPII signal. This is important because the absolute amount of RNAPII varies strongly across the genes. Without this normalization, the 5′ enrichment of RNAPII signal skews the apparent distribution of the modifications. Notably, the Ser7P depletion near the 5′ end of mRNAs we observed is not in agreement with prior observations of 5′ proximal enrichment of Ser7P (Mayer *et al*, 2010); however, in this analysis Ser7P enrichment was calculated relative to input DNA, whereas we calculated enrichment relative to the total RNAPII signal. In another study (Kim *et al*, 2010), where enrichment was calculated relative to total RNAPII, the authors noted that CTD Ser7P differs from Ser5P with the Ser5 5′peak relatively constant, whereas the profiles of Ser7P were variable over different genes consistent with our data.

A comparison of the unstable CUT class of lncRNAs with mRNAs revealed systematic differences in their RNAPII phosphorylation profiles. Both classes of transcripts had similar patterns of phosphorylation near their 5′ ends, but CUTs lacked the enrichment of

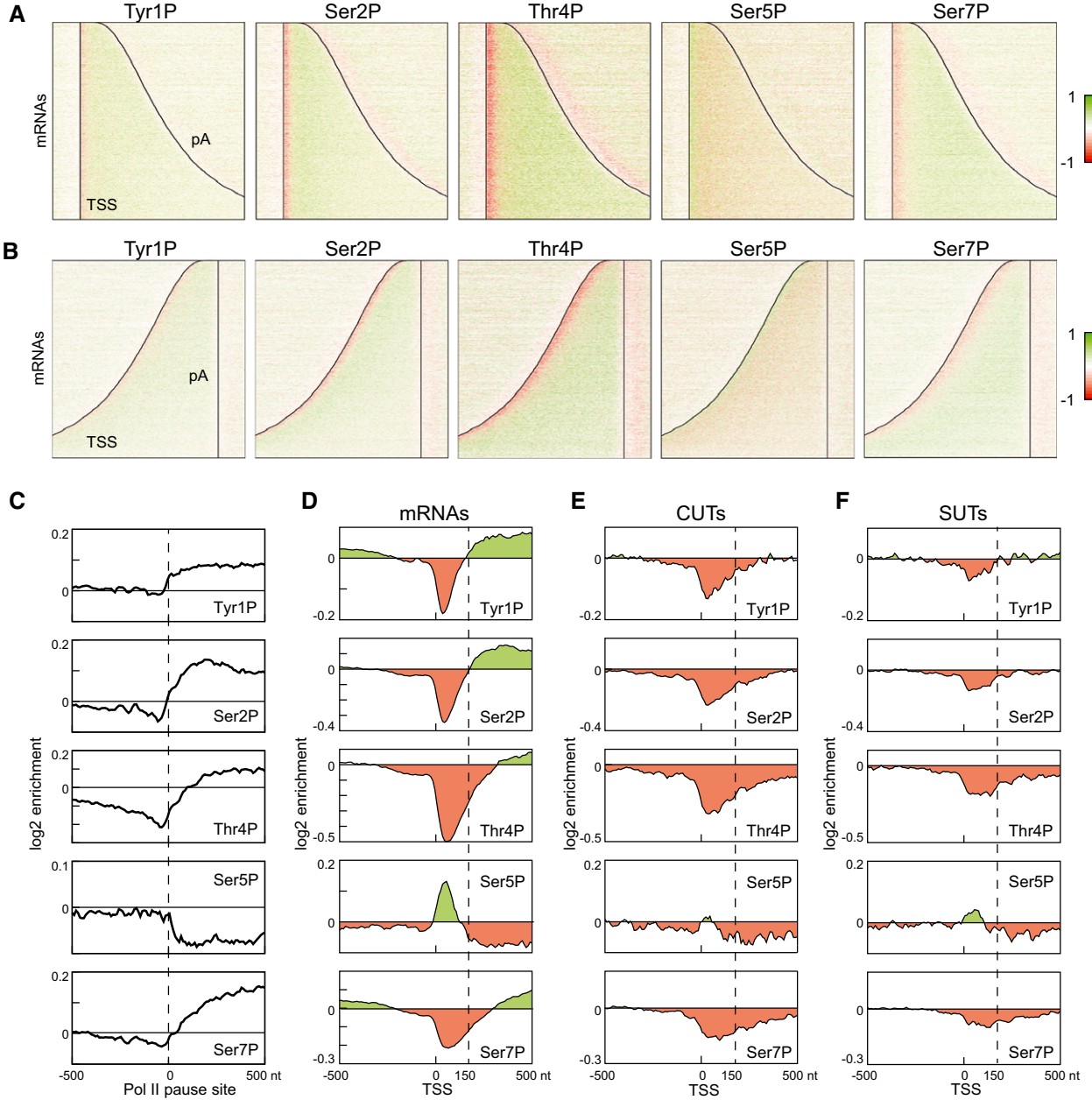

**Figure 3.  Profiles of RNAPII phosphorylation on mRNA and ncRNA genes can be generated using mCRAC.**

A   Distribution of RNAPII phosphorylation across protein-coding genes aligned at the TSS, as in Fig 1C as determined by mCRAC analyses on Rpo21-HTP. Red color indicates depletion, and green color indicates enrichment of phosphorylation relative to total RNAPII.

B   As panel (A) but with genes aligned at the polyadenylation site.

C   Metagene analysis of RNAPII phosphorylation enrichment relative to the 5,000 strongest RNAPII pause sites in mRNA genes, identified by NET-Seq (Churchman & Weissman, 2011).

D   Metagene analysis of RNAPII phosphorylation enrichment relative to transcription start sites, calculated for all mRNA genes. The TSS-proximal 150 nt region, where Ser5P is enriched and Ser2P and Tyr1P are depleted, is indicated with a dashed line.

E   Metagene analysis of RNAPII phosphorylation on CUTs as for (D).

F   Metagene analysis of RNAPII phosphorylation on SUTs as for (D).

Ser2P, Thr4P, and Ser7P seen in the middle, and toward the 3′ end, of most mRNAs (Figs 3D–F and EV4). As a result, CUTs show a global depletion of Ser2P, Ser7P, and most significantly, Thr4P. Analysis of the initial 500 nt of mRNAs, CUTs, and SUTs that are greater than 500 nt in length (Fig EV4B) confirmed that depletion of

the phosphorylation states associated with productive elongation is not due to the shorter length of CUTs. The more stable SUT class of ncRNA showed an intermediate pattern of modification (Fig 3F). Statistical analyses (Fig EV4) confirmed that differences in modification patterns between mRNAs, CUTs, and SUTs are significant

     

(Wilcoxon test with Bonferroni correction, $P < 0.01$ indicated by lines above the boxes on the boxplot) when compared over the entire transcripts (Fig EV4A) or over the first 500 nt of transcripts greater than 500 nt in length (Fig EV4B). Details of the analysis are listed in Table EV1.

## Generation of a hidden Markov model for RNAPII phosphorylation states

The mCRAC analyses generated high-resolution data for five different RNAPII modifications across thousands of transcription units from several different classes of RNA. To facilitate the extraction of biological insights from these complex data, we developed an eight-state, strand-specific hidden Markov model (HMM) that integrates the information from all RNAPII phosphorylation datasets. The model allowed us to partition each strand of the genome into segments characterized by recurrent patterns of RNAPII phosphorylation (Fig 4).

The HMM has only one parameter, the number $K$ of possible states, whose value has to be selected *a priori*. We tried different values of $K$ (from 3 to 15 states) and each time evaluated the data fit using the mean squared error (MSE) (see Materials and Methods). The MSE typically decreases as $K$ increases, as more complex models allow a better fit to the data. In this case there were inflection points at 6, 8, and 10 states (Fig EV5A). Analyses of models with 6, 8, or 10 gave qualitatively similar results (see below). We chose to perform most analyses using the HMM with 8 states, as it gave a good tradeoff between model fit and cost in terms of extra parameters to estimate. To interpret the segmentation returned by the HMM, we analyzed the profiles of states along mRNA transcripts.

A metagene analysis of the 8-state model revealed two initiation states (I1 and I2), an early elongation state (EE), three mid-/late elongation states (E1, E2, E3), a low phosphorylation state (L) enriched toward both ends of protein-coding genes, and a "noise" state (N), which was usually associated with non-transcribed regions of chromosomes (Fig 4A and B). These states showed a clear overall progression from I1 => EE => E1 on many genes (Fig 4A). The "emission matrix" showing the distribution of predicted phosphorylation patterns for each state is shown in Fig 4C. As expected, initiation states were characterized by intermediate (I1) to high (I2) levels of Ser5P and low levels of Ser2P and Thr4P. Consistent with our analysis of phosphorylation profiles, the early elongation state featured low Ser5P and Thr4P, with high Tyr1P, Ser2P, and Ser7P, whereas mid-/late elongation states had low Ser5P, and various combinations of high Tyr1P, Ser2P, Thr4P, and Ser7P. In particular, the major elongation state (E1) was associated with an elevated level of Thr4P. This may be related to the observation that the 5′ depletion of Thr4P extends further 3′ than that of Ser2P. In consequence, Thr4P levels increase at the location of state E1 rather than the early elongation state (EE). State L featured low levels of all phosphorylations (particularly Ser5P and Thr4P). Transition sites were strikingly reproducible between two replicate datasets (Fig EV5C).

RNAPII occupancy appeared to be sensitive to the presence of nucleosomes (Fig 1E). It therefore seemed possible that the locations of state transitions might be influenced by nucleosome positions, which are highly organized on most yeast genes (Jiang & Pugh, 2009). Comparing the locations of the end of the state I1 and the start of state EE to the reported nucleosome boundaries revealed enrichment for state I1 ending at or close to the 3′ boundary of nucleosome 1 and for state EE commencing at or close to the 5′ boundary of nucleosome 2 (Fig 4D and E). Comparison of all state changes with nucleosome boundaries (Fig EV5D) confirmed the enrichment of state I1 across the first nucleosome and the enrichment of state EE across nucleosomes 2 and 3. State E1 then increased over subsequent nucleosomes.

## The presence of an intron is associated with displacement of phosphorylation state boundaries

Pre-mRNA splicing has been reported to influence RNAPII phosphorylation in yeast (Alexander *et al*, 2010; Chathoth *et al*, 2014) and we therefore compared phosphorylation state boundaries in protein-coding genes lacking an intron (Fig 4F) with short exon 1 (< 100 nt) regions (Fig 4G) or with long (> 100 nt) exon 1 regions (Fig 4H). In Figs 4, EV5, and EV6, the data along each gene have been assigned to ten bins. This allows features to be aligned in genes of different lengths, although at the cost of reduced resolution. On intron-containing genes, initiation state I1 extended further 3′ relative to intronless genes, with a peak around the 5′ splice site. The elongation states EE and E1 were also displaced further 3′, presumably as a consequence of the extended I1 region. Notably, the same major conclusions could be reached from 6-state and 10-state HMM models (Fig EV6). This demonstrates that our conclusions hold independently of the specific implementation of the model. In comparison with the 8-state model, the 6-state model lacks a distinct early elongation (EE) state, probably because the complexity is too low, whereas the in 10-state model two initiation states have approximately equal frequencies.

Comparison of intron-containing and intronless genes that were aligned without binning gave similar results (Fig EV7). Alignment by the TSS indicated that the transition from state I1 to states EE and then E1 occurs further 3′ on spliced genes (Fig EV7A and B). Alignment of intron-containing genes at the 5′ splice site (5′SS) (Fig EV7C) showed that the maximum enrichment of state I1 coincided with the 5′ end of the intron, with 3′ displacement of the elongation states EE and E1. We conclude that the presence of an intron leads to changes in the transition sites for RNAPII modification patterns, upstream of the intronic regions. It was recently reported that splicing defects can lead to RNAPII accumulation over the exon 1 regions of endogenous genes (Chathoth *et al*, 2014), supporting this model.

Unexpectedly, alignment of pre-mRNAs at the 3′ splice site (3′ SS) showed specific depletion of the major elongation state E1 at this position, whereas states E2 and E3 appeared unaffected (Fig EV7C and data not shown). This indicates that the 3′ intron boundary is associated with distinct changes in RNAPII modification and also that the different elongation states are not fully interchangeable. The loss of state E1 at the 3′SS prompted us to examine the RNAPII phosphorylation data for these sites (Fig EV7D). This confirmed that specific changes in RNAPII phosphorylation are associated with 3′ splice sites. In particular, Ser7P was elevated at the 3′SS, with depletion of Ser2P and Thr4P. The depletion of Thr4P is the main factor in the loss of state E1 at the 3′SS. Whether these changes primarily reflect altered rates of phosphorylation or dephosphorylation remains to be determined.

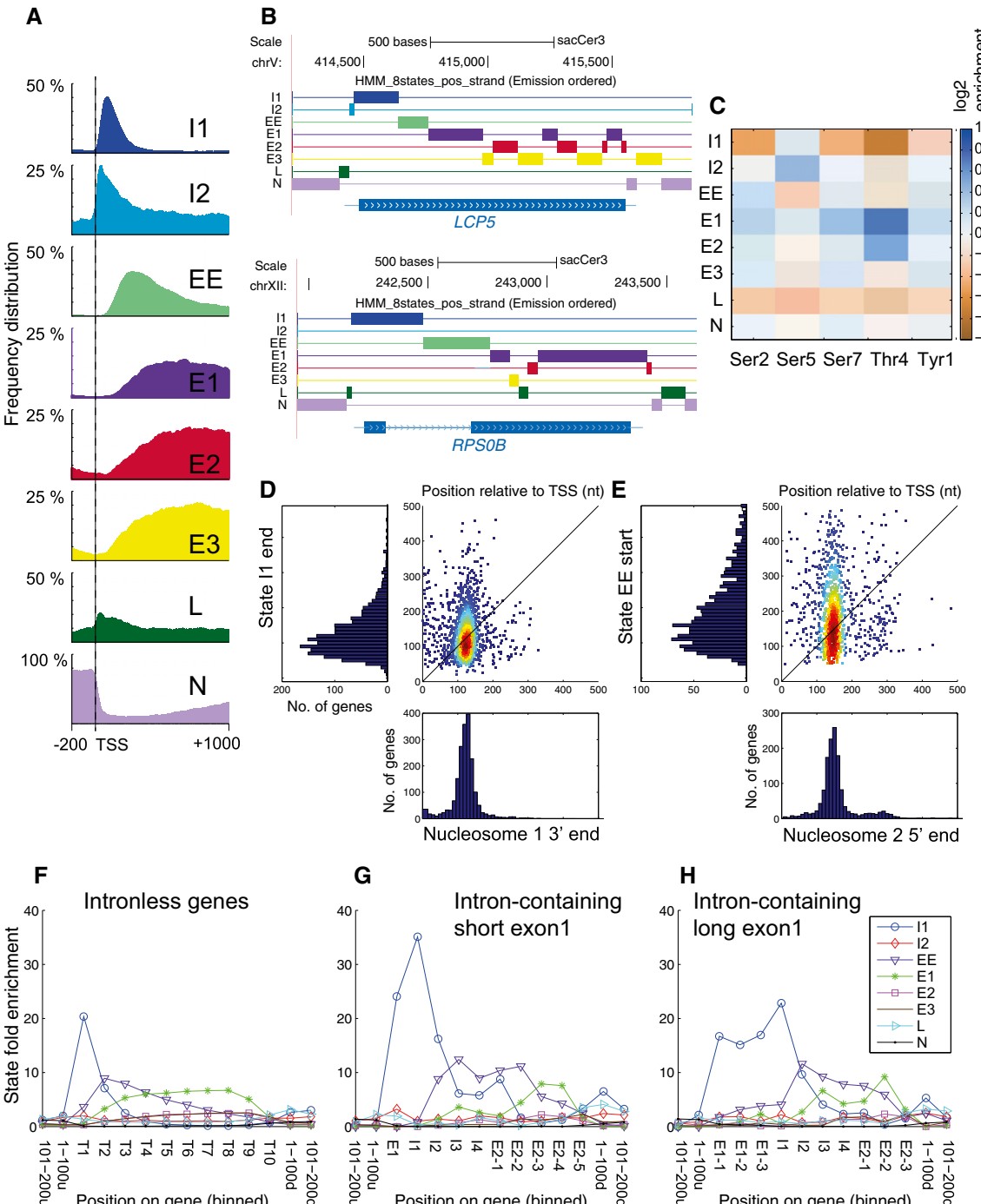

**Figure 4.    HMM analyses of phosphorylation state distributions.**

A    Metagene analysis of frequency distribution for each state on all protein-coding genes. See expanded view for analyses of replicate datasets and statistical analyses.

B    Genome browser views showing the distribution of the 8 states in the HMM over an unspliced gene (*LCP5*) and a spliced gene (*RPS0B*).

C    Learned emission matrix of the 8-state HMM. Each row shows the average log-enrichment levels of the different phosphorylated forms of Rpo21 over total RNAPII in one of the states.

D    Comparison of the locations of the 3′ end of initiation state I1 with the 3′ boundary of nucleosome 1.

E    Comparison of the locations of the 5′ end of early elongation state EE with the 5′ boundary of nucleosome 2.

F–H    The presence of an intron is associated with displacement of phosphorylation state boundaries. The graphs show state fold enrichment for each state over protein-coding genes lacking an intron (F), containing short exon 1 (< 100 nt) regions (G) or long (> 100 nt) exon 1 regions (H). For each panel, the length of each gene has been divided into ten bins to allow the combination of genes with different lengths. T, Transcript; E1, Exon 1; I, Intron; E2, Exon 2. Intron-containing genes in yeast are generally highly expressed, and the top quartile of intronless genes was therefore taken for comparison.

## State changes are altered on lncRNAs

We compared the representation of different HMM states on genes encoding mRNAs or the SUT and CUT classes of lncRNA (Fig 5A and B). To reduce potential effects of differences in expression levels on RNAPII modification patterns, we took the upper expression quartile of SUTs (211 genes), based on RNAPII crosslinking frequency, and identified sets of protein-coding genes (213 genes) and CUTs (211 genes) with matched expression levels. The mRNAs, CUTs, and SUTs showed similar levels of enrichment of the initiation state I1 close to the TSS. However, in comparison with mRNAs, the CUTs, and to a lesser extent SUTs, lack the early elongation state EE and the elongation states, particularly E1, confirming our findings that phosphorylation states associated with elongation were depleted on CUTs relative to length-matched mRNAs (Fig EV4). We conclude that on CUTs RNAPII largely fails to transition from the initiation states to the early elongation state EE and elongation states E1, E2, and E3.

On genes encoding CUTs, transcription termination is believed to be mediated by the Nrd1/Nab3/Sen1 (NNS) complex, which is proposed to directly recruit the TRAMP and exosome complexes to degrade or process the newly synthesized RNA (Arigo *et al*, 2006; Heo *et al*, 2013; Schulz *et al*, 2013; Tudek *et al*, 2014; Webb *et al*, 2014; Fasken *et al*, 2015). The CTD-interacting domain (CID) of Nrd1 preferentially binds CTD heptads with Ser5P modification *in vitro* (Vasiljeva *et al*, 2008). This led to the proposal that NNS termination and degradation of CUTs reflect their short lengths and the bias toward Ser5P modification close to the transcription initiation site. However, inspection of the data presented here indicates that on most mRNA genes RNAPII transition out of the Ser5P-associated initiation states into the early elongation state before reaching the median length of CUTs (377 nt) (Fig 5C), whereas these transitions do not generally occur on CUTs. These observations suggest that CUTs are short because RNAPII remains in a termination-prone, initiation state, the reverse of the previously proposed causality (Fig 6).

# Discussion

We report the development and application of mCRAC for high-resolution, strand-specific profiling of the five known types of RNAPII phosphorylation across the yeast transcriptome.

Comparison of RNAPII density over the + 50–150 and + 450–500 regions (on genes longer than 800 nt) indicated a reduction of 40–50% over this interval. Analysis of RNAPII position by the unrelated NET-seq protocol also revealed elevated polymerase density in the 5′ regions of protein-coding genes (Churchman & Weissman, 2011), indicating that this enrichment does not reflect differences in crosslinking efficiency. Moreover, high levels of promoter-proximal binding was seen for all of the key nuclear RNA surveillance factors tested; the exosome nuclease complex (Rrp44, Rrp6) and its major cofactor, the TRAMP nuclear polyadenylation complex (Trf4, Air2) (this work), as well as the NNS transcription termination and surveillance complex (Nab3), as previously reported (Creamer *et al*, 2011; Webb *et al*, 2014; Holmes *et al*, 2015). Notably, the 3′ ends of short TSS-associated ncRNA transcripts, termed sense CUTs (S CUTs), were previously mapped to the same region (Neil *et al*, 2009). In contrast, the mRNA export factor Mex67, which is

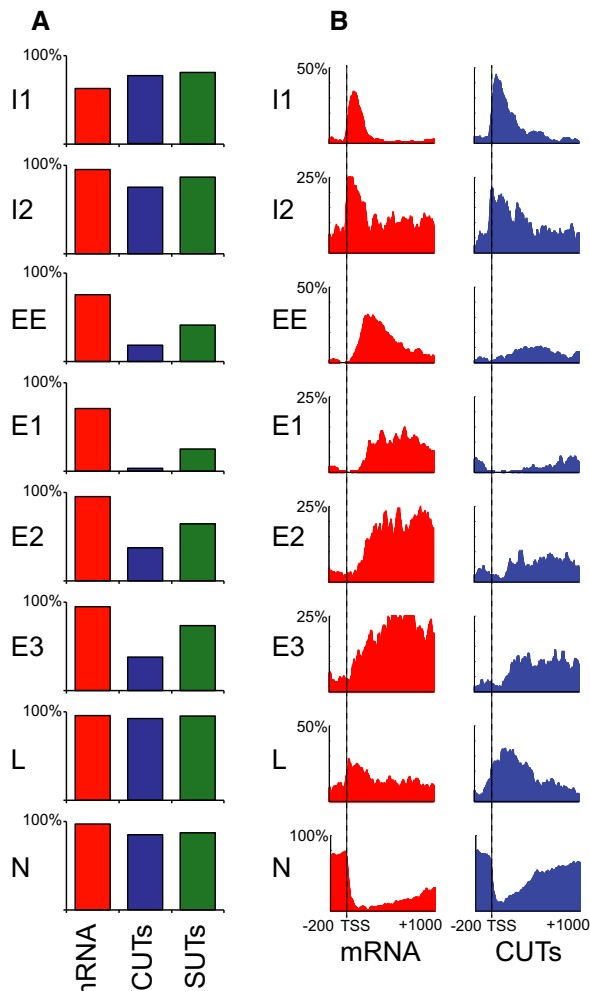

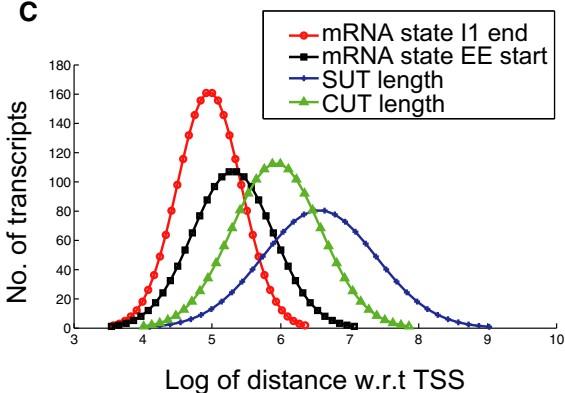

**Figure 5. Metagene HMM analyses of phosphorylation states on protein-coding and ncRNA genes.**

A Comparison of state frequencies over expression-matched protein-coding mRNAs, SUTs, and CUTs.

B Comparison of state distributions over expression-matched protein-coding mRNAs and CUTs as in A.

C The failure of lncRNAs to exit initiation state I1 is not a consequence of short length. The curves show the positions at which RNAPII exits initiation state I1 (red curve) and enters early elongation state EE (black curve), relative to the lengths of CUTs (green line) and SUTs (blue line).

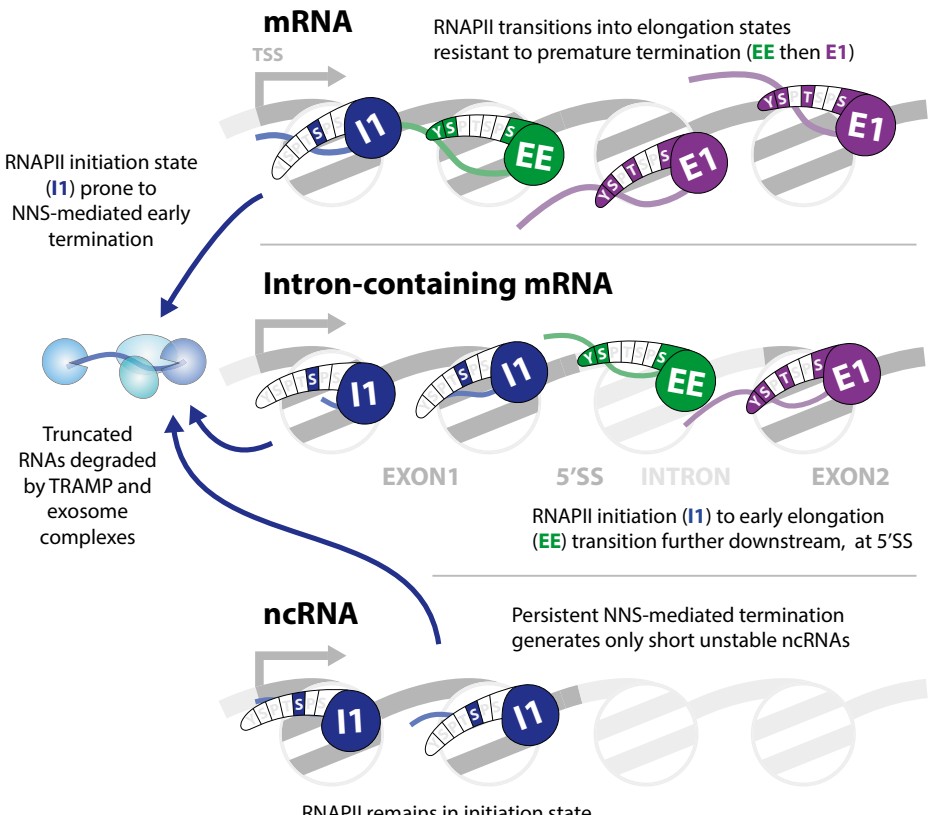

**Figure 6.  Model comparing the state transitions on coding and non-coding RNA transcripts.**
Yeast genes typically have a well-ordered nucleosome close to the transcription start site. On both mRNA and ncRNA genes, RNAPII generally in an initiation state (I1 and I2; shown as I1) while traversing the first nucleosome. This state favors recruitment of nuclear RNA surveillance machinery, including the NNS termination complex and the TRAMP–exosome degradation system. On mRNAs, RNAPII then transitions into the early elongation state (EE) followed by the major elongation states (E1 to E3; shown as E1). On intron-containing mRNAs, the transition from initiation of early elongation states is displaced further 3′. On ncRNAs, the initiation state persists and the failure to transition to an elongation states (EE and E1) favors termination and transcript degradation.

under-represented on ncRNAs relative to mRNAs (Tuck & Tollervey, 2013), was depleted from the TSS-proximal region.

These data are consistent with the model that short transcripts are generated by early termination, but are targets for rapid degradation by the exosome. In human cells, the generation of short, promoter-proximal ncRNAs has also been reported, termed PROMPTs, PASR, TSSa-RNAs, or tiRNAs (Kapranov *et al*, 2007; Preker *et al*, 2008, 2011; Seila *et al*, 2008; Jacquier, 2009; Taft *et al*, 2010). The accumulation of these relatively short ncRNAs is reported to correlate with cancer outcomes, supporting their functional significance (Zovoilis *et al*, 2014). The exosome also degrades ncRNAs generated from enhancer regions of human genes (Pefanis *et al*, 2015), suggesting that these may follow a similar degradation pathway.

Dramatic reductions in both total RNAPII occupancy and phosphorylation were also seen immediately upstream of the polyadenylation site. A similar reduction in RNAPII occupancy was seen in NETseq, showing that this does not reflect alterations in crosslinking efficiency. In contrast, this was not reported in previous ChIP analyses. It is possible that the relatively higher ChIP signals might be due to a combination of antisense transcription, which is seen in this region of most protein-coding genes (Fig 1), and transcript-free RNAPII immediately downstream of the poly(A) site. Notably, recent analyses in human cells have revealed the presence of a

transcription elongation checkpoint upstream of the poly(A) site (Laitem *et al*, 2015). We speculate that following satisfaction of a similar checkpoint, yeast RNAPII moves rapidly through the poly(A) region, leading to low CRAC and NETseq signals. Dramatically reduced RNAPII phosphorylation was also observed in this region, when expressed as a fraction of total RNAPII. We therefore propose that passage through the checkpoint is accompanied by RNAPII CTD dephosphorylation. Consistent with this model, a requirement for Tyr1 dephosphorylation prior to polyadenylation was recently demonstrated in yeast (Schreieck *et al*, 2014).

### Features potentially distinguishing mRNAs and lncRNAs

The synthesis of strikingly large numbers of lncRNAs has been reported in human cells and all other eukaryotes examined in detail. Many of these are rapidly degraded by the nuclear RNA surveillance system, particularly the TRAMP and exosome complexes, which are highly conserved between yeast and humans. In contrast, the lncRNAs represent a very diverse set of transcripts that show poor evolutionary conservation. This has made it difficult to discern common or conserved features in the lncRNA population, which might serve as markers that target them to the surveillance machinery. However, specific targeting of lncRNAs might not be required, if the TSS RNAPII

is intrinsically prone to transcription termination coupled to degrada-tion of the nascent transcript (Buratowski, 2009; Jensen *et al*, 2013; Tuck & Tollervey, 2013). We postulate that the generation of short, highly unstable RNAs may therefore be a default fate. This state of RNAPII is characterized by a distinct set of CTD modifications that can be recognized as a distinct initiation state by the HMM.

The Nrd1 component of the (NNS) complex preferentially binds the Ser5-modified RNAPII CTD *in vitro*, via its CTD-interacting domain (CID) (Vasiljeva *et al*, 2008). The NNS complex promotes transcription termination and recruits the TRAMP nuclear surveil-lance complex to the transcript, leading to its degradation by the exosome (Thiebaut *et al*, 2006; Tudek *et al*, 2014). On genes encoding the highly unstable CUT class of ncRNAs (Wyers *et al*, 2005), RNAPII predominantly failed to transition out of the Ser5P-rich, initiation state I1. This was seen to a lesser extent on SUT ncRNAs, which are more stable than the CUTs but are still largely targeted for degradation by the nuclear surveillance pathway (David *et al*, 2006; Tuck & Tollervey, 2013). In addition to differences in stability, CUTs (median length 377 nt) are generally shorter than SUTs (median length 745 nt) and mRNAs (median length 1,521 nt). Importantly, however, as shown in Fig 5C, the median lengths of CUTs and SUTs are markedly longer than the median locations of the transition from initiation to elongation state on protein-coding genes. Together, these observa-tions support the model shown in Fig 6. On both coding and non-coding RNAs, RNAPII is largely in initiation state I1 in the TSS region and is prone to termination and transcript degradation. On protein-coding genes, RNAPII transitions out of state I1 after transcrip-tion of a median of 122 nt, generally into the early elongation state EE and then elongation state E1. We propose that the failure of RNAPII on CUTs to transition out of state I1 underlies both the short length and high instability of these ncRNAs. Notably, previous attempts to define a specific instability element within CUTs lead to the suggestion that multiple, separable instability determinants must be present in each CUT (Thiebaut *et al*, 2006). Our model indicates an alternative explanation, that instability reflects interactions with the transcribing polymerase. Enhanced levels of Ser5-modified RNAPII were also seen adjacent to the 5′SS and at transcription pause sites predicted from the NETseq analyses. We speculate that in both cases RNAPII may be placed in a state that favors NNS recruitment, transcription termination, and rapid transcript degradation.

Peaks in RNAPII density seen here and in NETseq data showed a striking 150 nt periodicity that agrees well with the arrays of nucleo-somes formed on most yeast genes (Jiang & Pugh, 2009; Weiner *et al*, 2010; Churchman & Weissman, 2011). Notably, peak RNAPII binding was seen around the nucleosome centers, suggesting that nucleosomes may not generally be displaced by transcription, but the polymerase moves more slowly around the nucleosomes. This situation clearly has the potential to facilitate interactions between modified histones, or histone-binding proteins, and the transcribing polymerase or associated factors. Apparent correlations were observed between the locations of state changes in RNAPII and the positions of the first two nucleosomes, although it remains to be established whether these are causal. Consistent with this, the presence of H3K4me3 on the promoter-proximal nucleosome is associated with NNS recruitment (Terzi *et al*, 2011).

The antibodies used for CTD analyses generally recognize a single phosphorylated amino acid, although in some cases recogni-tion is modulated by other specific phosphorylated residues in the

same or the adjacent heptad (see Eick & Geyer, 2013). However, we cannot currently determine which, or how many, of the 26 heptad repeats are actually modified *in vivo* on specific transcripts. If the modifications are dispersed across the repeats, then competing phosphorylation sites seems unlikely to be a major problem. In any event, the effects of competition between modifications on antibody binding are unlikely to alter the locations of transitions in the HMM, although they might affect the mechanistic interpretation in terms of precise protein interactions at these sites.

Finally, many other protein factors bind to the phospho-CTD, including the TREX pre-mRNA packaging complex, and factors required for pre-mRNA splicing and polyadenylation (Meinel *et al*, 2013). The segmentation of the genome by the HMM should also facilitate understanding of the interactions between these factors and the transcribing polymerase.

# Materials and Methods

### Strains and yeast culture

*Saccharomyces cerevisiae* strain BY4741 was used for all experi-ments. Prior to crosslinking, cultures were grown to $OD_{600}$ 0.5 at 30°C in standard, minimal YNB medium with 2% glucose, supple-mented with complete supplement mix minus tryptophan (all from MP Biomedicals).

To assess the distribution of RNAPII, an Rpo21-HTP construct (C-terminal, tripartite tag; His6-TEV protease cleavage site–protein A) was integrated at the *RPO21* chromosomal locus under the control of the endogenous $P_{RPO21}$ promoter. Rpo21-HTP was the only source of Rpo21 and growth of the resulting strains was indis-tinguishable from the parental wild type (BY4741), indicating that the tagged construct is functional. To assess the distribution of RNAPII, an Rpo21-HTP construct (C-terminal, tripartite tag; His6-TEV protease cleavage site–protein A) was integrated at the *RPO21* chromosomal locus under the control of the endogenous $P_{RPO21}$ promoter. Rpo21-HTP was the only source of Rpo21 and growth of the resulting strains was indistinguishable from the parental wild type (BY4741), indicating that the tagged construct is functional. CRAC analyses on Rrp6 and Trf4 were performed using strains expressing previously described Rrp6-HTP and Trf4-HTP constructs (Schneider *et al*, 2012; Tuck & Tollervey, 2013). Strains expressing Air2-HTP and Rrp44-HTP constructs were constructed by integrating the tag at the chromosomal locus, as described above.

### mCRAC analysis

UV crosslinking at 254 nm was performed on actively growing Rpo21-HTP cells as well as cells carrying an untagged allele of Rpo21 as a negative control. Rpo21-HTP was recovered from the lysate by binding of the protein A tag to an IgG column and elution by TEV cleavage, essentially as described (Granneman *et al*, 2011; Schneider *et al*, 2012). Differentially phosphorylated forms of RNAPII were then separated by immunoprecipitation using anti-bodies purchased from Chromotek directed against the CTD carrying Tyr1P (3D12), Ser2P (3E10), Thr4P (6D7), Ser5P (3E8), or Ser7P (4E12) bound to sheep anti-rat IgG Dynabeads, or a no antibody control with sheep anti-rat IgG Dynabeads alone. RNA–protein

complexes were subjected to partial RNase degradation, denatured by the addition of guanidinium–HCl to 4 M and bound to a nickel column via the His6 tag on Rpo21. Linkers were added to the Rpo21-associated RNAs on the nickel column and the bound protein–RNA complexes were eluted with imidazole and gel purified by SDS–PAGE (Fig EV1). Following proteinase K digestion, RNAs were identified by reverse transcription and PCR amplification, followed by Illumina sequencing. Illumina sequencing was performed by Edinburgh Genomics and Source Biosystems. CRAC analyses on Rrp6-HTP, Rrp44-HTP, Air2-HTP, and Trf4-HTP were performed essentially as described (Granneman et al, 2011; Schneider et al, 2012).

The replicate RNAPII CRAC datasets were compared to all annotated transcripts. The average proportions of transcripts captured were as follows: at least 1 read, 95% CUTs, 93% SUTs, 99% mRNAs; at least 10 reads, 54% CUTs, 60% SUTs, 97% mRNAs. It is not expected all genes will be active under the specific growth condition used.

## Sequence data analysis

The FASTX-Toolkit (http://hannonlab.cshl.edu/fastx_toolkit/) was used for the preprocessing of Illumina sequencing data: Reads were trimmed using fastx-clipper to remove the 3′-linker sequence and quality-filtered using fastq_quality_trimmer (-t 30) and fastq_quality_filter (-q 23 -p 100). Preprocessed reads were demultiplexed using pyCRAC (Webb et al, 2014) and index and random barcode sequences from the 5′ ends of reads were removed. Reads were mapped to the yeast genome (sacCer3, downloaded from the Saccharomyces Genome Database) using novoalign (−r Random −s 1). Alternatively, mapping reads with TopHat gave very similar results. To remove potential PCR duplicates, reads mapped to the same start position in the genome and sharing the same barcode were collapsed, as described in Tuck and Tollervey (2013).

Bedtools (Quinlan & Hall, 2010) and custom AWK and Perl scripts were used for downstream analyses. For calculation of coverage pie charts, reads were uniquely assigned to RNA biotypes using a hierarchical procedure, starting with the most abundant biotypes. Transcript annotations were from the Saccharomyces Genome Database and (Xu et al, 2009) (mRNAs, CUTs, and SUTs). Where appropriate, transcript coordinates were converted to the sacCer3 genome assembly using liftOver. Coverage around genomic features (metagene analyses and 2D heatmaps) was plotted using custom R and gnuplot scripts. To calculate the enrichment of phosphorylated RNAPII signal relative to total RNAPII signal, normalized coverage (reads per million) was calculated at each position along the genome for all datasets and averaged between replicate datasets. Log2 enrichment was calculated after the addition of 5 pseudocounts to both the numerator and denominator; this number was selected because it represented about one read in the dataset with the lowest coverage. Pab1 and Xrn1 CRAC data were from Tuck and Tollervey (2013), and NET-Seq coordinates were from Churchman and Weissman (2011).

## Modeling the mCRAC data with a hidden Markov model

A multivariate hidden Markov model (HMM) was used to model jointly the different RNAPII phosphorylation datasets. The HMM was learned from the log-enrichment values (i.e., log2(phosphorylated Pol II + 5/total Pol II + 5)), where a pseudocount of 5 was added to each genomic region to avoid numerical instabilities. The entire segmentation data can be found in Dataset EV2.

An HMM is a probabilistic model, which explains a sequence of observations $\{x_1, x_2, \ldots, x_N\}$ through an unobserved (latent) sequence of discrete states. In our case, each variable $x_i$ is a five-dimensional vector describing the $Tyr_1$, $Ser_2$, $Thr_4$, $Ser_5$, and $Ser_7$ phosphorylation log-enrichment levels over total RNAPII in the $i$th 20 nt-bin along the genome. Since mCRAC provides strand-specific information, the two strands of the genome were simply concatenated and modeled together, avoiding the need for more specialized computational tools (Zacher et al, 2014). The architecture of a HMM consists of a Markov chain of latent variables $\{z_1, z_2, \ldots, z_N\}$, where each variable $z_n$ can be in one of $K$ states. The state of $z_n$ only depends on the state of the previous latent variable $z_{n-1}$, and each observation $x_n$ is conditioned on the state of the corresponding latent variable $z_n$. The model is characterized by two sets of parameters:

- The *transition* probabilities $P(z_n = j \mid z_{n-1} = i)$, describing the probability for $z_n$ to be in state $j$ given that $z_{n-1}$ is in state i $(i, j = 1, \ldots, K)$;
- The *emission* probabilities $P(x_n \mid z_n)$, describing the probability distribution of the observed variable $x_n$, conditioned on the state of $z_n$. In our model, we assume Gaussian emission probabilities: $P(x_n \mid z_n = i) \sim \mathcal{N}(x_n \mid \mu_i, \Sigma_i), i = 1, \ldots, K$. Therefore, the parameters associated with the emission probabilities are a set of 5 mean values and 20 covariance values for each state.

The values of the different model parameters, that is, the transition probabilities, and the means $\mu_i$ and covariances $\Sigma_i$ of the Gaussian distributions, are learned by running an expectation maximization algorithm. The values of the transition probabilities are initialized randomly from a uniform distribution, while the means $\mu_i$ are initialized by applying the $K$-means clustering algorithm to the data and the covariances $\Sigma_i$ are initialized to the covariance matrices of the obtained clusters. The most probable sequence of states for each strand is identified using the Viterbi algorithm. The MATLAB scripts we used to learn the HMM can be found in Code EV1.

The HMM has only one parameter, the number $K$ of possible states, whose value has to be selected *a priori*. We tried different values of $K$ (from 3 to 15 states) and each time evaluated the data fit using the mean squared error (MSE):

$$\text{MSE} = \frac{1}{10N} \sum_{s=1}^{2} \sum_{n=1}^{N} \| x_n^s - \mu_{z_n}^s \|^2,$$

where superscript $s$ refers to the strand.

The MSE typically decreases as $K$ increases, as more complex models allow a better fit to the data, but appeared to level off after 8 states, yielding diminishing returns in terms of model fit for the same cost in terms of extra parameters to estimate. We chose therefore to retain the HMM with 8 states for further analyses as it gave a good tradeoff between model complexity and model fit. The transition and emission matrices of the 8-state model are shown in Figs 4C and EV6A, respectively. As shown in Fig EV6, qualitatively similar conclusions could be drawn from models with 6 or 10 states.

## Reproducibility of the HMM results

We ran the HMM algorithm with 10 different random parameter initializations, and each time checked the similarity between the new obtained sequence of states and the one we used for our analysis. Both for the positive and negative strands, the percentage of 20 nt-windows that are predicted to be in the same state in both sequences is 87% ($\pm$ 5%) on average (to be compared to the random baseline of 12.5%).

We also compared the sequences of states obtained from two independent mCRAC datasets. The Jaccard index is a commonly used way to assess the similarity of two sample sets. It computes the ratio A/B, where A (resp. B) is the size of the intersection (resp. union) of the two sets. Table EV2 shows the Jaccard index for each state $i$, measuring the similarity of the sets of 20 nt-windows that are predicted to be in state $i$ in the two HMM sequences, respectively (with positive and negative genomic strands presented separately). The Jaccard indices that are shown in Table EV2 are significantly high, each corresponding to a *P*-value of $< 10^{-4}$ (computed using 1,000 random permutations of one of the sequences), indicating a high similarity between the two replicates. The two state distributions across the first quartile of protein-coding genes are also highly similar (Fig EV5C).

## State enrichment analysis

Given the predicted sequence of hidden states along the genome, the enrichment of a state $i$ in a window of interest is computed as the ratio $n_{i,obs}/n_{i,exp}$, where $n_{i,obs}$ (resp. $n_{i,exp}$) is the observed (resp. expected) number of states $i$ within this window. The expected number of states $i$ is computed from the total number of states $i$ that are observed along the genome and assuming a uniform distribution.

## Data availability

All sequences generated during this work have been deposited with GEO and can be accessed under the accession number GSE69676. Additional datasets used for Fig 2 are available from GEO. Mex67 CRAC data: Tuck AC, Tollervey D. A transcriptome-wide atlas of RNP composition reveals diverse classes of mRNAs and lncRNAs. Accession number: GSE46742. CUT 3′ end data: Neil *et al* (2009). Accession number: GSE25132. Nucleosome position coordinates were obtained from Additional data file 1 of Jiang and Pugh (2009).

**Expanded View** for this article is available online.

## Acknowledgements

This work was supported by the Wellcome Trust through a Principal Research Fellowship to D.T. (077248). A.T. is supported by the Wellcome Trust (103977). CD-F was supported by a FEBS Long-Term Fellowship. G.K. was supported by the Wellcome Trust (097383) and by the Medical Research Council. V.A.H.-T. and G.S. acknowledge support from the European Research Council under grant MLCS 306999. Work in the Wellcome Trust Centre for Cell Biology is supported by Wellcome Trust core funding (092076).

## Author contributions

LM and DT conceived the project and designed experiments; LM, EP, and CD-F constructed strains and performed CRAC experiments; VAH-T, AT, RL, GS, CD-F, and GK performed bioinformatic analysis; GS, GK, and DT wrote the paper.

## Conflict of interest

The authors declare that they have no conflicts of interest.

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
