## [Review Process File · Molecular Systems Biology]

Strand-specific, high-resolution mapping of modified RNA polymerase II

Laura Milligan, Vân A Huynh-Thu, Clémentine Delan-Forino, Alex Tuck, Elisabeth Petfalski, Rodrigo Lombrana, Guido Sanguinetti, Grzegorz Kudla and David Tollervey

Corresponding authors: David Tollervey, Grzegorz Kudla and Guido Sanguinetti, The University of Edinburgh

Review timeline:

Submission date:	08 February 2016
Editorial Decision:	04 March 2016
Revision received:	04 April 2016
Editorial Decision:	28 April 2016
Revision received:	13 May 2016
Accepted:	18 May 2016

Editor: Maria Polychronidou

Transaction Report:

1st Editorial Decision

04 March 2016

Thank you again for submitting your work to Molecular Systems Biology. We have now heard back from the three referees who agreed to evaluate your study. As you will see below, the reviewers think that the presented methodology and data are a valuable contribution to the field. However, they raise a number of concerns, which should be carefully addressed in a revision of the manuscript. The reviewers' recommendations are rather clear so there is no need to repeat the points listed below.

REFeree COMMENTS

Reviewer #1:

Comments on the paper (MSB-16-6869 Milligan et al.,)

This study describes a new methodological approach to analyze nascent RNA associated with different RNAPII CTD modifications at single nucleotide resolution in yeast; called the mCRAC method. Combined with this mCRAC method and the Hidden Markov Model (HMM), they show 6, 8 and 10 model states of nascent RNA and define several transition points. Interestingly noncoding RNA display a transition failure between some elongation states that distinguishes them from protein-coding RNA. Overall the data is expertly analyzed and this allows many previously

predicted transcriptional events to be visualized. However there are few wholly new discoveries made. Several points that need to be clarified are listed below.

Specific comments

1. mCRAC is the first method to describe CTD-modification specific genome-wide nascent RNA profiles in budding yeast. However, this has already been reported in human cells (Nojima et al., Cell 2015) and should be appropriately cited.
2. Meta-profiles show average gene patterns. A few specific gene profiles should also be shown especially for Figure 1C and 1E.
3. Figure S1. The SDS PAGE gel image require sizing controls. Are these bands RNAPII ? If the two panels are duplicates why do they differ?
4. Page 6. They call exon-exon junction EE. This is confusing as EE is also used for HMM analysis. A different term for the spliced product is needed such as ExEx?
5. Page 9. They cite NET-seq with the wrong reference. Churchman's group calls their nascent RNA-sequence method human NET-seq (Mayer et al., 2015). However they do not isolate RNAPII complexes so their NET-seq detects the 3'end of chromatin-bound RNA, but does not detect an RNAPII protected fragment.
6. Figure 3. Again they should add individual gene examples.
7. Figure 4C. They need to explain why the E1 state has significant T4P RNAPII?
8. On Page 12, second paragraph, last sentence. They show S7P is increased at 3'ss with a depletion of S2P. However T4P is also depleted. They should explain why S2P and T4P levels decreased at 3'ss.
9. Technical question. Why does the mCRAC method needs UV-crosslinking? Other methods do not use crosslinking (Churchman and Weissman 2011 and Nojima et al., 2015) as the interaction between RNAPII and nascent RNA is stable. They also mentioned there are no significant differences on their profiles between yeast NET-seq and mCRAC methods. What are the reasons for using crosslinking in the mCRAC method?

Reviewer #2:

The authors present an elegant approach to measure the modifications of RNAPII in a strand-specific manner. The study recapitulates what is known about CTD modification patterns at mRNAs. For the first time these modifications can be studied in a sound way at CUTs and SUTs, which is problematic with ChIP protocols since these transcripts are often located in antisense direction of mRNAs. Analysis based on a hidden Markov model indicates that CUTs generally do not leave the state of initiation, probably due to their short length and early termination.

Overall, this study provides a very valuable resource for the yeast transcriptional community and raises interesting hypotheses. However, I have three major points, which should be addressed before the manuscript is suitable for publication:

Major points:

CRAC of RNAPII and its modifications makes it possible to disentangle strand-specific binding, which is especially important for studying transcripts as CUTs and SUTs which are frequently located in antisense direction to mRNAs. It would be great if the authors could quantify how CRAC improves over ChIP-Seq in that matter. This could be done for instance by comparing correlations of the non-strand-specific ChIP with CUT expression and the strand-specific CRAC signals with CUT expression in overlapping regions.

Figure 3E:

The authors propose a model in which CUTs do not enter productive elongation, which is characterized by Ser2P, and therefore exit transcription and get degraded. The difficulty with this claim is to disentangle causes from consequences. The lower enrichment of Ser2P for CUTS in Figure 3E could be due to the shorter length of CUTs, since Ser2P levels have been shown to increase with the distance to TSSs (Mayer et al., 2010). Thus mRNAs are expected to show higher (average) Ser2P levels because they are longer. To control for potential confounding due to

differences in length distribution, it would be better to consider only e.g. the first 200, 300, or 500nt of transcripts with length \geq 200, 300, 500nt, respectively.

Figure 5C:

The authors suggest that CUTs do not enter productive elongation because mRNAs typically enter the corresponding HMM state (EE) before the median length of CUTs. It is difficult to make such claim using the HMM, because this statistical model classifies states at a given genomic position also based on signal from neighboring positions. The authors should come back to the original CRAC data to support this claim. Distributions of relevant modifications as function of distance to TSS for mRNA versus CUTs should be provided.

Minor point:

Page 9. There is an apparent contradiction in the discussion of the literature about Ser7P. Kim et al 2010 is cited for two contradictory facts: i) 'prior observations of 5' proximal enrichment of Ser7P' and ii) 'In another study (Kim et al, 2010), where enrichment was calculated relative to total RNAPII, the authors noted that CTD Ser7P differs from Ser5P [...]'.
 Figure 4A,B: colors do not match. The magenta and light pink state in B are not shown in A.

Reviewer #3:

In this manuscript, Milligan et al develop a new method (mCRAC) to map phospho-isoforms of RNA polymerase II on RNA, in a strand-specific manner across the transcriptome. They combine implementation and development of the mCRAC method with development of a new computational approach to analyze the high degree of complexity present in transcriptome-wide datasets such as these.

Key findings include periodic spacing of RNAPII on transcripts, with peaks coinciding with nucleosome positions; the definition of different RNAP II states and how these are distributed across RNAs; and the persistence of initiation states on short ncRNAs including CUTS, and on intron-containing genes until after the first exon. Ser5P is enriched (and other phosphorylation marks are depleted) near the TSS. All phosphorylations are depleted close to the polyA site.

Overall, the study is well-conceived and of considerable interest to the field. The results broadly agree with several recent publications which mapped elongating RNAPII on transcripts and the distribution of RNAP II phosphorylations across the CTD. This study is unique both in methodology and results, going beyond these other studies. I recommend that it is published after addressing the following points.

Main points:

- 1) Since this paper describes a new method, more details should be included. For example, the authors should mention more explicitly any negative controls (no antibody control). Are control experiments performed with non RNA-binding proteins? What is the percentage of the RNAP II transcriptome that is captured in a typical experiment?
- 2) The figures need substantial improvements. Many axes are not labeled and there are often no scales shown (Fig 1B, 2, 3 (A has min/max but no numbers), 4, S5, S6). In cases where data were normalized, this is not indicated. It should be clear from the figure and legend how the data were normalised to a relative scale.
- 3) The discussion of nucleosome positioning influencing the elongation rate was compelling, and it was interesting that the I1 - EE state transition occurred at the first nucleosome boundary (Fig 4D,E). However, given that many transcripts presumably extend over a second nucleosome boundary (from nucleosome 2 to 3), the authors don't seem to comment at all on this. Even if there

are no further state changes observed, or perhaps if this analysis is prevented by poor coverage of transcripts that extend over a second nucleosome boundary, it should at least be mentioned.

4) The poor coverage of RNAP II-associated 3' UTR sequences (i.e. after the stop codon) was surprising (but also observed in other studies). Could this be related to nucleosome positioning relative to the 3' end (leading to rapid transcription of this region, p14)? Are sequenced fragments from the 3'UTR less likely to be uniquely mapped due to lower complexity?

5) The number of states for HMM was evaluated by the MSE. The authors state that this levels off after 8 states. This should be plotted and shown as a supplemental figure.

6) The discussion figure 6 should be expanded to present a graphical model for splice-site boundary events.

Minor points:

-Abstract, 2nd sentence: make it clear that this method maps RNAP II on RNA

-p3, first paragraph: referencing is sparse

-p4 RNPII instead of RNAPII

-p5, errors in "We propose that close the transcription start site" and "On gene encoding unstable"

-p7, RNAII instead of RNAPII

-p7-8/Fig 2, It is not clear whether the mapping of all surveillance factors is from this work or from other published work. This is listed in the figure legend but should be more explicit. Methods are only given for Rpo21. Show Hrp1 and Nab2 distributions as well for comparison.

-Fig S2 - similarly are the Pab1 and Xrn1 data from this work or previous work?

p9, relatively

p10, top: reference to Figure 3D should be for Figure 3E.

p11, top: The authors state the RNAPII elongation rates appear to be sensitive to the presence of nucleosomes. It would be more appropriate to state that RNAPII density is sensitive to the presence of nucleosomes since elongation rates have not been measured.

p11, reference to Fig S6 in middle of page should be to S4 and S5.

-p25 Fig1A and S4 legends state that this is a CLAMP protocol. Presumably the authors mean mCRAC.

-p31 There is an error in the legend for S6A.

-Figure 1B the colors are difficult to differentiate.

-Fig S1 should be consistent with Fig 1A. TEV cleavage isn't indicated on the Rpo21 construct halfway down the page. Define Nab.

-Fig S5, labels for panels C and D are missing

1st Revision - authors' response

04 April 2016

Reviewer #1:

Comments on the paper (MSB-16-6869 Milligan et al.)

This study describes a new methodological approach to analyze nascent RNA associated with different RNAPII CTD modifications at single nucleotide resolution in yeast; called the mCRAC method. Combined with this mCRAC method and the Hidden Markov Model (HMM), they show 6, 8 and 10 model states of nascent RNA and define several transition points. Interestingly noncoding RNA display a transition failure between some elongation states that distinguishes them from protein-coding RNA. Overall the data is expertly analyzed and this allows many previously predicted transcriptional events to be visualized. However there are few wholly new discoveries made. Several points that need to be clarified are listed below.

Specific comments

1. *mCRAC is the first method to describe CTD-modification specific genome-wide nascent RNA profiles in budding yeast. However, this has already been reported in human cells (Nojima et al., Cell 2015) and should be appropriately cited.*

We have altered the text in the Introduction (p3) to make this clear.

2. *Meta-profiles show average gene patterns. A few specific gene profiles should also be shown especially for Figure 1C and 1E.*

We have added images for 4 individual genes showing the RNAPII distribution and reported nucleosome boundary positions as new Figure EV2.

3. *Figure S1. The SDS PAGE gel image require sizing controls. Are these bands RNAPII ? If the two panels are duplicates why do they differ?*

This labeling was performed solely to identify the gel region that should be excised for further analysis, which is done by placing the gel on top of the autoradiograph. The labeling is combined with the 5' phosphorylation of the partially degraded RNA using unlabeled ATP, which is required prior to linker ligation, and was not performed with careful quantitation. The different intensities do accurately reflect relative RNAPII recovery and do not affect the outcome of the experiment. For simplicity, we now show only one gel in the revised MS.

4. *Page 6. They call exon-exon junction EE. This is confusing as EE is also used for HMM analysis. A different term for the spliced product is needed such as ExEx?*

Good point - we have changed the nomenclature as proposed.

5. *Page 9. They cite NET-seq with the wrong reference. Churchman's group calls their nascent RNA-sequence method human NET-seq (Mayer et al., 2015). However they do not isolate RNAPII complexes so their NET-seq detects the 3' end of chromatin-bound RNA, but does not detect an RNAPII protected fragment.*

We have altered the text to correct this error.

6. *Figure 3. Again they should add individual gene examples.*

As noted above, new Figure EV2 shows reported nucleosome boundary positions.

7. *Figure 4C. They need to explain why the E1 state has significant T4P RNAPII?*

As the referee notes, state E1 is associated with an elevated level of Thr4P. This may be related to the observation that the 5' depletion of Thr4P extends further 3' than that of Ser2P. In consequence, Thr4P levels increase at the location of the major, late elongation state (E1) rather than the early elongation state (EE). We have altered the text to include these points (p11).

8. *On Page 12, second paragraph, last sentence. They show S7P is increased at 3'ss with a depletion of S2P. However T4P is also depleted. They should explain why S2P and T4P levels decreased at 3'ss.*

The referee makes a very useful point. The depletion of Thr4P is actually the main factor in the loss of state E1 at the 3'SS. Mechanistically, we are currently unable to determine whether these changes primarily reflect altered rates of phosphorylation or dephosphorylation. We have altered the text to include these points (p12).

9. *Technical question. Why does the mCRAC method needs UV-crosslinking? Other methods do not use crosslinking (Churchman and Weissman 2011 and Nojima et al., 2015) as the interaction between RNAPII and nascent RNA is stable. They also mentioned there are no significant differences on their profiles between yeast NET-seq and mCRAC methods. What are the reasons for*

using crosslinking in the mCRAC method?

The crosslinking step in mCRAC method allows very stringent purification under denaturing conditions. It is a potential concern with NET-seq that RNAs recovered might, for example, be associated with RNA processing factors that are in turn bound to the polymerase.

Reviewer #2:

The authors present an elegant approach to measure the modifications of RNAPII in a strand-specific manner. The study recapitulates what is known about CTD modification patterns at mRNAs. For the first time these modifications can be studied in a sound way at CUTs and SUTs, which is problematic with ChIP protocols since these transcripts are often located in antisense direction of mRNAs. Analysis based on a hidden Markov model indicates that CUTs generally do not leave the state of initiation, probably due to their short length and early termination.

Overall, this study provides a very valuable resource for the yeast transcriptional community and raises interesting hypotheses. However, I have three major points, which should be addressed before the manuscript is suitable for publication:

Major points:

CRAC of RNAPII and its modifications makes it possible to disentangle strand-specific binding, which is especially important for studying transcripts as CUTs and SUTs which are frequently located in antisense direction to mRNAs. It would be great if the authors could quantify how CRAC improves over ChIP-Seq in that matter. This could be done for instance by comparing correlations of the non-strand-specific ChIP with CUT expression and the strand-specific CRAC signals with CUT expression in overlapping regions.

The problem with globally quantifying CUTs relative to overlapping and antisense transcripts is that we do not really have quantitative datasets with which to compare our CRAC data. In RNA seq or microarrays the CUTs are scarcely detectable in total RNA at steady state. They were mapped in strains lacking exosome components. However, it is unclear exactly what was the degree of stabilization conferred by these mutations, which are also highly pleiotropic. However, the contribution of CUTs to total transcription is readily visualized at individual sites. As an example of this, the *PHO84* gene has a well-characterized antisense transcript that is prominently visible in the CRAC data show in the new Fig. EV2.

Figure 3E:

The authors propose a model in which CUTs do not enter productive elongation, which is characterized by Ser2P, and therefore exit transcription and get degraded. The difficulty with this claim is to disentangle causes from consequences. The lower enrichment of Ser2P for CUTS in Figure 3E could be due to the shorter length of CUTs, since Ser2P levels have been shown to increase with the distance to TSSs (Mayer et al., 2010). Thus mRNAs are expected to show higher (average) Ser2P levels because they are longer. To control for potential confounding due to differences in length distribution, it would be better to consider only e.g. the first 200, 300, or 500nt of transcripts with length $\geq 200, 300, 500$ nt, respectively.

We have included additional panel F in the revised version of Figure 3, showing that Ser2P is depleted in the region 1-500 in ncRNAs, but not in mRNAs. This is also mentioned in the revised text (p10, para 3).

Figure 5C:

The authors suggest that CUTs do not enter productive elongation because mRNAs typically enter

the corresponding HMM state (EE) before the median length of CUTs. It is difficult to make such claim using the HMM, because this statistical model classifies states at a given genomic position also based on signal from neighboring positions. The authors should come back to the original CRAC data to support this claim. Distributions of relevant modifications as function of distance to TSS for mRNA versus CUTs should be provided.

We have introduced new panels in Fig. EV3B-D to address this point and mention these data in the text (p10 and p13).

Minor point:

Page 9. There is an apparent contradiction in the discussion of the literature about Ser7P. Kim et al 2010 is cited for two contradictory facts: i) 'prior observations of 5' proximal enrichment of Ser7P' and ii) 'In another study (Kim et al, 2010), where enrichment was calculated relative to total RNAPII, the authors noted that CTD Ser7P differs from Ser5P [...]'

We thank the referee; the reference to Kim et al. was accidentally included twice. We have changed the text (p10).

Figure 4A,B: colors do not match. The magenta and light pink state in B are not shown in A.

We have changed the colors in the resubmitted figure. These now match – and also correspond with the genome browser views in new Fig. EV2.

Reviewer #3:

In this manuscript, Milligan et al develop a new method (mCRAC) to map phospho-isoforms of RNA polymerase II on RNA, in a strand-specific manner across the transcriptome. They combine implementation and development of the mCRAC method with development of a new computational approach to analyze the high degree of complexity present in transcriptome-wide datasets such as these.

Key findings include periodic spacing of RNAPII on transcripts, with peaks coinciding with nucleosome positions; the definition of different RNAP II states and how these are distributed across RNAs; and the persistence of initiation states on short ncRNAs including CUTS, and on intron-containing genes until after the first exon. Ser5P is enriched (and other phosphorylation marks are depleted) near the TSS. All phosphorylations are depleted close to the polyA site.

Overall, the study is well-conceived and of considerable interest to the field. The results broadly agree with several recent publications which mapped elongating RNAPII on transcripts and the distribution of RNAP II phosphorylations across the CTD. This study is unique both in methodology and results, going beyond these other studies. I recommend that it is published after addressing the following points.

Main points:

1) Since this paper describes a new method, more details should be included. For example, the authors should mention more explicitly any negative controls (no antibody control). Are control experiments performed with non RNA-binding proteins?

Details have been added to Experimental Procedures (p19, para 3).

What is the percentage of the RNAP II transcriptome that is captured in a typical experiment?

We have determined the fraction of the annotated transcripts that are recovered in the RNAPII CRAC data. Details have been added to Experimental Procedures (p20, para 1).

2) *The figures need substantial improvements. Many axes are not labeled and there are often no scales shown*

Fig 1B

This is a piechart. It may be that the referee intended Fig. 1C, to which we have added color scale bars.

Fig. 2.

Axes have been labeled and normalization is described in the legend.

Fig. 3 (A has min/max but no numbers)

A color scale bar has been added to the revised figure.

Additional labels have been added to

Fig. 4A, C, D, E, F, G H.

Fig. EV5B, C

Fig.EV6A, B, C, D

In cases where data were normalized, this is not indicated. It should be clear from the figure and legend how the data were normalized to a relative scale.

Information on normalization has been added to the legend of figure 2. Details of normalization for the HMM are included in the Experimental Procedures.

3) *The discussion of nucleosome positioning influencing the elongation rate was compelling, and it was interesting that the II - EE state transition occurred at the first nucleosome boundary (Fig 4D,E). However, given that many transcripts presumably extend over a second nucleosome boundary (from nucleosome 2 to 3), the authors don't seem to comment at all on this. Even if there are no further state changes observed, or perhaps if this analysis is prevented by poor coverage of transcripts that extend over a second nucleosome boundary, it should at least be mentioned.*

We have looked at these boundaries and have included a graph in the revised version of Fig. EV5.

4) *The poor coverage of RNAP II-associated 3' UTR sequences (i.e. after the stop codon) was surprising (but also observed in other studies). Could this be related to nucleosome positioning relative to the 3' end (leading to rapid transcription of this region, p14)?*

This is an interesting idea. We plotted the positions of nucleosomes around the 3' end and there is indeed a striking pattern. We have added this graph to the revised Figure EV3 and point out this correlation in the text. However, the causality remains unclear.

Are sequenced fragments from the 3'UTR less likely to be uniquely mapped due to lower complexity?

We did not select uniquely mapped reads. Reads mapped to more than one location are randomly allocated. Moreover, the complexity of the 3' UTR region does not appear to be low enough for substantial mapping problems.

5) *The number of states for HMM was evaluated by the MSE. The authors state that this levels off after 8 states. This should be plotted and shown as a supplemental figure.*

This graph has been included in the revised version of Fig. EV5.

6) *The discussion figure 6 should be expanded to present a graphical model for splice-site boundary events.*

We have included this in the revised version of Fig. 6.

Minor points: -

-Abstract, 2nd sentence: make it clear that this method maps RNAP II on RNA
The text has been changed (p2).

-p3, first paragraph: referencing is sparse
Additional references have been included.

-p4 RNPII instead of RNAPII
Corrected

-p5, errors in "We propose that close the transcription start site" and "On gene encoding unstable"
Corrected

-p7, RNAPII instead of RNAPII
Corrected

-p7-8/Fig 2, It is not clear whether the mapping of all surveillance factors is from this work or from other published work. This is listed in the figure legend but should be more explicit.
We have also placed this information in the revised Results section (p8).

Methods are only given for Rpo21.
We have included descriptions of the Rrp44, Rrp6, Trf4 and Air2 CRAC analyses in the Experimental Procedures (p18).

Show Hrp1 and Nab2 distributions as well for comparison.
We have included these results in the revised version of Fig. 2.

-Fig S2 - similarly are the Pab1 and Xrn1 data from this work or previous work?
The data are from Tuck and Tollervy (2013). This information was included in the revised figure legend and Experimental Procedures.

p9, relatively
Corrected

p10, top: reference to Figure 3D should be for Figure 3E.
Corrected

p11, top: The authors state the RNAPII elongation rates appear to be sensitive to the presence of nucleosomes. It would be more appropriate to state that RNAPII density is sensitive to the presence of nucleosomes since elongation rates have not been measured.
Corrected

p11, reference to Fig S6 in middle of page should be to S4 and S5.
Corrected

-p25 Fig1A and S4 legends state that this is a CLAMP protocol. Presumably the authors mean mCRAC.
Corrected

-p31 There is an error in the legend for S6A.

Corrected

-Figure 1B the colors are difficult to differentiate.

The colors have been changed in the revised figure

-Fig S1 should be consistent with Fig 1A. TEV cleavage isn't indicated on the Rpo21 construct halfway down the page. Define NAb.

The figure has been changed and NAb is defined in the legend and Experimental Procedures.

-Fig S5, labels for panels C and D are missing

The labels have been added.

2nd Editorial Decision

28 April 2016

Thank you for submitting your revised manuscript to Molecular Systems Biology. We have now heard back from the three referees who, as you will see below, think that most of their major concerns have been satisfactorily addressed. However, they still list some remaining concerns, which we would ask you to address in a revision.

REFeree COMMENTS

Reviewer #1:

Generally the revised ms addresses all of our comments.

However comments 2 and 6 (Reviewer 1) could do with further ms modification:

The revised ms shows individual profiles in Figure EV2. However these images are very small so that it is hard to see the differences in different CTD modified Pol II profiles. Furthermore a cutoff value is used on the Y-axis. Ideally a different scale should be employed so that peak differences are clearly visible. For example, in the PHO84 gene, the mCRAC signals look essentially flat, especially with S5P.

Reviewer #2:

I have one concern left (over two comments). The point is not demonstrated yet. I have a suggestion that may help the authors to make it.

Reviewer's original point:

Figure 3E:

The authors propose a model in which CUTs do not enter productive elongation, which is characterized by Ser2P, and therefore exit transcription and get degraded. The difficulty with this claim is to disentangle causes from consequences. The lower enrichment of Ser2P for CUTs in Figure 3E could be due to the shorter length of CUTs, since Ser2P levels have been shown to increase with the distance to TSSs (Mayer et al., 2010). Thus mRNAs are expected to show higher (average) Ser2P levels because they are longer. To control for potential confounding due to differences in length distribution, it would be better to consider only e.g. the first 200, 300, or 500nt of transcripts with length \geq 200, 300, 500nt, respectively."

Authors: We have included additional panel F in the revised version of Figure 3, showing that Ser2P is depleted in the region 1-500 in ncRNAs, but not in mRNAs. This is also mentioned in the revised text (p10, para 3).

Reviewer: The figures 3E-F do not allow comparisons of the different classes. The boxplots should be organized by marks showing the three transcript classes side-by-side than having transcript classes in distinct panels on top of each other. Actually, boxplots of 3F show that over the 1-500 nt mRNA have lower Ser2P (a bit), as well as much lower T4P and lower S7P than along the whole gene (i.e. compared to 3E). Hence, mRNA profiles in the 1-500nt are more resembling the SUTs and CUTS profile in the same regions. The text should state that the relative depletion of these modifications is seen for mRNA in the 1-500n t region and is exaggerated for CUTs and SUTs. Moreover, this claim should be supported by a test for statistical significance (e.g. two-sided Wilcoxon test). Furthermore, the text claims that SUTs have distinct profiles than CUTs ("The more stable SUT class of ncRNA showed an intermediate pattern of modification (Figure 3E)"). This claim should be supported by statistical testing for the 500 nt region (comparing boxplots 3F for SUTs vs. CUTs). It would be if at all significant of very little effect.

Reviewer's original point:

Figure 5C:

The authors suggest that CUTs do not enter productive elongation because mRNAs typically enter the corresponding HMM state (EE) before the median length of CUTs. It is difficult to make such claim using the HMM, because this statistical model classifies states at a given genomic position also based on signal from neighboring positions. The authors should come back to the original CRAC data to support this claim. Distributions of relevant modifications as function of distance to TSS for mRNA versus CUTs should be provided.

Authors: We have introduced new panels in Fig. EV3B-D to address this point and mention these data in the text (p10 and p13).

Reviewer: These plots are great and could be swapped with 3E-F in my opinion. For proper comparison, they should be done however for transcripts longer than 500 nt. These plots also suggest that performing the statistical comparisons analysis mentioned above separately for the regions 1-150nt and for the regions 150-500nt would give more signal since these two regions have often opposite patterns and thus cancel each other when pooled.

Reviewer #3:

The authors have addressed all of my concerns in their revised manuscript. The text and figures are much improved.

I noticed a couple minor errors:

pg 10, second paragraph: "Analysis of the initial 500 nt of mRNAs, CUTs and SUT that are " should be SUTs instead of SUT.

Page 30, figure 3 legend: "The graph below each panel shows a metagene analysis of RNAPII phosphorylation enrichment for all mRNA genes." There is no graph below each panel.

Page 33, Figure EV6 legend: instead of upper, middle and lower, I think it should be left, middle and right graphs.

2nd Revision - authors' response

13 May 2016

Reviewer #1:

Generally the revised ms addresses all of our comments.

However comments 2 and 6 (Reviewer 1) could do with further ms modification:

The revised ms shows individual profiles in Figure EV2. However these images are very small so

that it is hard to see the differences in different CTD modified Pol II profiles. Furthermore a cutoff value is used on the Y-axis. Ideally a different scale should be employed so that peak differences are clearly visible. For example, in the PHO84 gene, the mCRAC signals look essentially flat, especially with S5P.

Authors: Figure EV2 has been converted into dataset EV1 and the figure size has been increased. We have altered the scale to remove the cutoff. In the case of PHO84, the minus strand (mRNA) signals appear to be clear in our version of the figure. On the plus strand, there are some phosphorylation signals that can be attributed to the antisense transcript. These signals are less pronounced, although the Ser5 enrichment around nt 23,900 is readily visible, and strong enough for the HMM model to call an initiation state.

Reviewer #2:

I have one concern left (over two comments). The point is not demonstrated yet. I have a suggestion that may help the authors to make it.

Reviewer's original point:

Figure 3E:

The authors propose a model in which CUTs do not enter productive elongation, which is characterized by Ser2P, and therefore exit transcription and get degraded. The difficulty with this claim is to disentangle causes from consequences. The lower enrichment of Ser2P for CUTs in Figure 3E could be due to the shorter length of CUTs, since Ser2P levels have been shown to increase with the distance to TSSs (Mayer et al., 2010). Thus mRNAs are expected to show higher (average) Ser2P levels because they are longer. To control for potential confounding due to differences in length distribution, it would be better to consider only e.g. the first 200, 300, or 500nt of transcripts with length $\geq 200, 300, 500$ nt, respectively."

Authors: We have included additional panel F in the revised version of Figure 3, showing that Ser2P is depleted in the region 1-500 in ncRNAs, but not in mRNAs. This is also mentioned in the revised text (p10, para 3).

Reviewer: The figures 3E-F do not allow comparisons of the different classes. The boxplots should be organized by marks showing the three transcript classes side-by-side than having transcript classes in distinct panels on top of each other. Actually, boxplots of 3F show that over the 1-500 nt mRNA have lower Ser2P (a bit), as well as much lower T4P and lower S7P than along the whole gene (i.e. compared to 3E). Hence, mRNA profiles in the 1-500nt are more resembling the SUTs and CUTS profile in the same regions. The text should state that the relative depletion of these modifications is seen for mRNA in the 1-500nt region and is exaggerated for CUTs and SUTs. Moreover, this claim should be supported by a test for statistical significance (e.g. two-sided Wilcoxon test). Furthermore, the text claims that SUTs have distinct profiles than CUTs ("The more stable SUT class of ncRNA showed an intermediate pattern of modification (Figure 3E)."). This claim should be supported by statistical testing for the 500 nt region (comparing boxplots 3F for SUTs vs. CUTs). It would be if at all significant of very little effect.

Authors: The box plots have been redrawn as requested and are now shown as Figures EV3B and EV3C. The statistical analyses are presented as Table EV1. Most tests are significant, also after correction for multiple testing (Wilcoxon test with Bonferroni correction, $p < 0.01$ indicated by lines above the boxes on the boxplot). The elongation-related modifications, Y1P, S2P, T4P and S7P, are significantly enriched on mRNAs relative to CUTs and SUTs, and on SUTs relative to CUTs. This pattern of enrichment is consistent with the relative stabilities of the three classes of transcripts, and we have updated the main text to highlight this point.

Reviewer's original point:

Figure 5C:

The authors suggest that CUTs do not enter productive elongation because mRNAs typically enter the corresponding HMM state (EE) before the median length of CUTs. It is difficult to make such claim using the HMM, because this statistical model classifies states at a given genomic position

also based on signal from neighboring positions. The authors should come back to the original CRAC data to support this claim. Distributions of relevant modifications as function of distance to TSS for mRNA versus CUTs should be provided.

Authors: We have introduced new panels in Fig. EV3B-D to address this point and mention these data in the text (p10 and p13).

Reviewer: These plots are great and could be swapped with 3E-F in my opinion. For proper comparison, they should be done however for transcripts longer than 500 nt. These plots also suggest that performing the statistical comparisons analysis mentioned above separately for the regions 1-150nt and for the regions 150-500nt would give more signal since these two regions have often opposite patterns and thus cancel each other when pooled.

Authors: The plots are now included in Figure 3. As shown in Table EV1 and Figure EV3, the patterns of enrichment are very similar on full-length transcripts, and on regions 1-500 nt of transcripts of length >500 nt. For this reason, and to reduce noise by averaging a larger number of transcripts, we decided to show all transcripts rather than subsets of transcripts in Figures 3D-F.

Reviewer #3:

The authors have addressed all of my concerns in their revised manuscript. The text and figures are much improved.

I noticed a couple minor errors:

pg 10, second paragraph: "Analysis of the initial 500 nt of mRNAs, CUTs and SUT that are " should be SUTs instead of SUT.

Page 30, figure 3 legend: "The graph below each panel shows a metagene analysis of RNAPII phosphorylation enrichment for all mRNA genes." There is no graph below each panel.

Page 33, Figure EV6 legend: instead of upper, middle and lower, I think it should be left, middle and right graphs.

Authors: Corrected

Corresponding Author Name: Cees Dekker
Manuscript Number: MSB-15-6724R